# A novel fold for acyltransferase-3 (AT3) proteins provides a framework for transmembrane acyl-group transfer

Kahlan E Newman[1†], Sarah N Tindall[2†], Sophie L Mader[3], Syma Khalid[3]*, Gavin H Thomas[2]*, Marjan W Van Der Woude[4]*

[1]School of Chemistry, University of Southampton, Southampton, United Kingdom; [2]Department of Biology and the York Biomedical Research Institute, University of York, York, United Kingdom; [3]Department of Biochemistry, University of Oxford, Oxford, United Kingdom; [4]Hull York Medical School and the York Biomedical Research Institute, University of York, York, United Kingdom

*For correspondence:
Syma.khalid@bioch.ox.ac.uk (SK);
gavin.thomas@york.ac.uk (GHT);
marjan.vanderwoude@york.ac.uk
(MWVDW)

†These authors contributed
equally to this work

Competing interest: The authors
declare that no competing
interests exist.

Reviewing Editor: Qiang Cui,
Boston University, United States

**Abstract** Acylation of diverse carbohydrates occurs across all domains of life and can be catalysed by proteins with a membrane bound acyltransferase-3 (AT3) domain (PF01757). In bacteria, these proteins are essential in processes including symbiosis, resistance to viruses and antimicrobials, and biosynthesis of antibiotics, yet their structure and mechanism are largely unknown. In this study, evolutionary co-variance analysis was used to build a computational model of the structure of a bacterial O-antigen modifying acetyltransferase, OafB. The resulting structure exhibited a novel fold for the AT3 domain, which molecular dynamics simulations demonstrated is stable in the membrane. The AT3 domain contains 10 transmembrane helices arranged to form a large cytoplasmic cavity lined by residues known to be essential for function. Further molecular dynamics simulations support a model where the acyl-coA donor spans the membrane through accessing a pore created by movement of an important loop capping the inner cavity, enabling OafB to present the acetyl group close to the likely catalytic resides on the extracytoplasmic surface. Limited but important interactions with the fused SGNH domain in OafB are identified, and modelling suggests this domain is mobile and can both accept acyl-groups from the AT3 and then reach beyond the membrane to reach acceptor substrates. Together this new general model of AT3 function provides a framework for the development of inhibitors that could abrogate critical functions of bacterial pathogens.

## Editor's evaluation

By integrating a range of computational techniques, the authors made an important contribution by generating a structural model for the AT3 domain, which is predicted to adopt a new fold. The key features of the structural model are consistent with the activity of the enzyme as an acyltransferase, with a transmembrane channel that can accommodate an acyl-CoA donor, and an outer cavity formed with a second domain that can accommodate a nascent LPS molecule as substrate. Overall, the study is valuable as it will help stimulate specific experimental analyses that will further evaluate and improve the model for better mechanistic understanding of this class of enzymes.

## Introduction

Acyltransferase family 3 (*Acyl_transf_3,* AT3) domain (Interpro: IPR002656, PFAM: PF01757) containing proteins are found in all domains of life. These membrane-bound AT3 proteins are involved in acylation of a wide range of extracytoplasmic and surface bacterial polysaccharides (*Pearson et al., 2022*)

**eLife digest** The fatty membrane that surrounds cells is an essential feature of all living things. It is a selective barrier, only allowing certain substances to enter and exit the cell, and it contains the proteins and carbohydrates that the cell uses to interact with its environment. In bacteria, the carbohydrates on the outer side of the membrane can become 'tagged' or modified with small chemical entities which often prove useful for the cell. Acyl groups, for example, allow disease-causing bacteria to evade the immune system and contribute to infections persisting in the body.

As a rule, activated acyl groups are only found inside the cell, so they need to move across the membrane before they can be attached onto the carbohydrates at the surface. This transfer is performed by a group of proteins that sit within the membrane called the acyltransferase-3 (AT3) family. The structure of these proteins and the mechanism by which they facilitate membrane crossing have remained unclear.

Newman, Tindall et al. combined computational and structural modelling techniques with existing experimental data to establish how this family of proteins moves acyl groups across the membrane. They focused on OafB, an AT3 protein from the foodborne bacterial pathogen *Salmonella typhimurium*. The experimental data used by the team included information about which parts of OafB are necessary for this protein to acylate carbohydrates molecules.

In their experiments, Newman, Tindall et al. studied how different parts of OafB move, how they interact with the molecules that carry an acyl group to the membrane, and how the acyl group is then transferred to the carbohydrate acceptor. Their results suggest that AT3 family proteins have a central pore or hole, plugged by a loop. This loop moves and therefore 'unplug' the pore, resulting in the emergence of a channel across the membrane. This channel can accommodate the acyl-donating molecule, presenting the acyl group to the outer surface of the membrane where it can be transferred to the acceptor carbohydrate.

The AT3 family of proteins participates in many cellular processes involving the membrane, and a range of bacterial pathogens rely on these proteins to successfully infect human hosts. The results of Newman Tindall et al. could therefore be used across the biological sciences to provide more detailed understanding of the membrane, and to inform the design of drugs to fight bacterial diseases.

but are also important in Eukarya, for example, in the regulation of lifespan in *Caenorhabditis elegans* (**Vora et al., 2013**) and in *Drosophila* development (**Dzitoyeva et al., 2003**). In bacteria, where these proteins have been primarily studied, the resulting acylations have been shown to be involved in root nodulation (**Davis et al., 1988**), increase the efficacy of macrolide antibiotics (**Hara and Hutchinson, 1992**), conferring resistance to lysozyme (**Laaberki et al., 2011**), influencing bacteriophage sensitivity (**Teh et al., 2020**), and altering antibody recognition (**Davies et al., 2013**; **Broadbent et al., 2010**; **Kintz et al., 2017**). While AT3 domains most commonly exist as standalone proteins, there are many examples of AT3 domains fused with SGNH domains, with alanine racemase domains, and fusions to other domains also exist in Eukarya (see Pfam architectures for PF10757). AT3 domains are predicted to have 10 transmembrane helices (TMHs; **Allison and Verma, 2000**; **Bernard et al., 2011**; **Bohin, 2000**; **Bonnet et al., 2017**; **Bontemps-Gallo et al., 2016**; **Cogez et al., 2002**; **Corvera et al., 1999**; **Geno et al., 2017**; **Kajimura et al., 2006**; **Kintz et al., 2015**; **Lacroix et al., 1999**; **Menéndez et al., 2004**; **Moynihan and Clarke, 2011**; **Slauch et al., 1996**; **Spencer et al., 2017**; **Thanweer et al., 2008**; **Zou et al., 1999**; **Buendia et al., 1991**; **Bera et al., 2005**); however, despite the wide-ranging functions of this family of proteins, there are currently no models for their overall structure and only limited information on mechanism.

The overall understanding of AT3 proteins is mainly derived from studies of these proteins in the context of bacterial virulence through changes on the cell surface (**Vora et al., 2013**; **Dzitoyeva et al., 2003**). One example of this is the O-acetylation of the O-antigen of the lipopolysaccharide (LPS) and lipooligosaccharide by AT3 domain containing proteins present in the inner (cytoplasmic) membrane of Gram-negative bacteria, including species of *Neisseria* (**Kahler et al., 2006**), *Salmonella* (**Hellerqvist et al., 1969**; **Hellerqvist et al., 1971**), *Shigella* (**Clark et al., 1991**; **Sun et al., 2012**; **Verma et al., 1991**; **Knirel et al., 2014**; **Wang et al., 2014**; **Sun et al., 2014**), *Haemophilus* (**Fox et al., 2005**; **Yildirim et al., 2005**), *Burkholderia* (**Brett et al., 2011**; **Brett et al., 2003**; **Heiss et al., 2012**; **Heiss**

*et al., 2013*), and *Legionella* (*Zou et al., 1999*; *Kooistra et al., 2001*; *Wagner et al., 2007*). Specifically, residues in the O-antigen of the LPS (e.g. rhamnose, abequose) can be O-acetylated during LPS synthesis and before LPS export to the outer membrane. As outlined below, current evidence indicates this acetylation takes place in the periplasm, which is the compartment between the inner and outer membrane in these Gram-negative bacteria. This process is distinct from that leading to acylation of the lipid A tail of LPS. Acetylation of the O-antigen by AT3 family membrane proteins increases O-antigen heterogeneity, which can lead to an altered bacterial serotype and resistance to bacteriophage (*Hellerqvist et al., 1969*; *Hellerqvist et al., 1971*; *Clark et al., 1991*; *Sun et al., 2012*; *Verma et al., 1991*; *Knirel et al., 2014*). Some of these reactions require AT3-only proteins, others are carried out by proteins with an AT3 domain fused with a C-terminal SGNH domain. SGNH proteins are a family of proteins with hydrolase/transferase activity (*Akoh et al., 2004*).

A second important process in bacteria that can be mediated by an AT3 protein is O-acetylation of peptidoglycan that leads to resistance to both lysozyme and β-lactam antibiotics (*Bera et al., 2005*; *Aubry et al., 2011*). One such protein, OatA, is an AT3 protein with attached SGNH domain (AT3-SGNH), that acetylates the MurNAc residue in peptidoglycan (*Bera et al., 2005*; *Aubry et al., 2011*). Initially discovered in *Staphylococcus aureus* (*Bera et al., 2005*), OatA homologues have since been identified in many Gram-positive bacteria including *Listeria monocytogenes* (*Aubry et al., 2011*), *Lactococcus lactis* (*Cao et al., 2018*; *Veiga et al., 2007*), and *Streptococcus pneumoniae* (*Bonnet et al., 2017*; *Crisóstomo et al., 2006*; *Davis et al., 2008*). Furthermore, standalone AT3 proteins contribute an acyl group in the biosynthesis of macrolide antibiotics in *Streptomyces* species which increases antibiotic efficacy (*Hara and Hutchinson, 1992*; *Arisawa et al., 1995*; *Arisawa et al., 1994*; *Epp et al., 1989*). These selected examples (see *Pearson et al., 2022* for a recent review) illustrate that AT3 domain-containing proteins can O-acylate a diverse range of acceptor molecules in different contexts (e.g. O-antigen, peptidoglycan), that are involved in a range of highly relevant processes for bacterial pathogens and may be relevant for biotechnological applications.

Found in the cytoplasmic membrane, AT3 domains are highly hydrophobic integral membrane proteins predicted to contain 10 TMHs (*Allison and Verma, 2000*; *Bernard et al., 2011*; *Bohin, 2000*; *Bonnet et al., 2017*; *Bontemps-Gallo et al., 2016*; *Cogez et al., 2002*; *Corvera et al., 1999*; *Geno et al., 2017*; *Kajimura et al., 2006*; *Kintz et al., 2015*; *Lacroix et al., 1999*; *Menéndez et al., 2004*; *Moynihan and Clarke, 2011*; *Slauch et al., 1996*; *Spencer et al., 2017*; *Thanweer et al., 2008*; *Zou et al., 1999*; *Buendia et al., 1991*; *Bera et al., 2005*). While the majority of AT3 domains consist of only an AT3 domain (standalone AT3), many are AT3 domains with an additional TMH and C-terminal, periplasmic SGNH domain attached via a periplasmic linking region (AT3-SGNH) (*Meziane-Cherif et al., 2015*). *Thanweer et al., 2008* and *Jones et al., 2021* have performed PhoA-LacZα fusion analysis of the O-antigen acetyltransferase Oac from *Shigella flexneri* (a standalone AT3) and OatA from *S. aureus* (an AT3-SGNH), respectively. Analysis by *Thanweer et al., 2008* suggested that Oac has 10 TMH, with the N- and C-termini in the cytoplasm. This is consistent with fusion analysis of the O-antigen acetyltransferase OafB from *Salmonella* ser. Typhimurium (*Kintz et al., 2015*). The *Salmonella* O-antigen acetyltransferases OafB and OafA are both AT3-SGNH fusion proteins and predicted to have an 11th TMH to allow the fused SGNH domain to be located in the periplasm (*Kintz et al., 2015*; *Pearson et al., 2020*). In contrast, *Jones et al., 2021* proposed that OatA contains only 9 TMH with one large re-entrant loop; as an AT3-SGNH protein, OatA was also previously predicted to contain 11 TMH, orienting the SGNH domain in the periplasm (*Bernard et al., 2011*; *Bera et al., 2005*). In addition, a large cytoplasmic loop was described between TMH 8 and 9 of OatA, where Oac has only short cytoplasmic loops but a longer periplasmic loop between TMH3 and 4 (*Thanweer et al., 2008*; *Rajput and Verma, 2022*). Currently, this is the extent of the knowledge of the architecture of AT3 domains, and there are no published experimentally determined protein structures.

AT3 proteins are predicted to transfer acyl groups, including acetyl groups, from the cytoplasm across the cytoplasmic membrane to be transferred onto the extra-cytoplasmic acceptor molecule (*Laaberki et al., 2011*). However, the mechanism is currently largely unknown. The acetyl group has been proposed to be donated by cytoplasmic acetyl coenzyme A (acetyl-CoA)(*Slauch et al., 1996*). Three arginine residues on the cytoplasmic side of TMH1 and 3 have been identified as highly conserved in the PFAM HMM logo, and in both standalone AT3 and AT3-SGNH proteins these residues are essential for function (*Thanweer et al., 2008*; *Pearson et al., 2020*). Arginine residues have been shown to be involved in binding of the 3'-phosphate group of acetyl-CoA (*Wu and Hersh,*

*1995*) suggesting that these conserved residues may interact with the proposed acetyl group donor. Previous in vitro studies suggest that standalone AT3 proteins CmmA, MdmB, and Asm19 are able to utilise acetyl-CoA as an acetyl donor (*Hara and Hutchinson, 1992*; *Moss et al., 2002*; *García et al., 2011*). Importantly, *Jones et al., 2021* found OatA from *S. aureus* (an AT3-SGNH protein) are able to hydrolyse the acetyl group from acetyl-CoA and transfer it to the peptidoglycan-like acceptor substrate, effectively observing the entire catalytic cycle in vitro (*Jones et al., 2021*). Thus, the available data indicate that acetyl-CoA is the most likely acetyl group donor, but how this interacts with the AT3 domain or how the acetyl group can cross the membrane, is as yet unknown.

In addition, site-directed mutagenesis identified three conserved tyrosine residues which are required for function (*Jones et al., 2021*). One of these Tyr residues is predicted to be located in the periplasm and is proposed to be involved in transfer of the acyl group from the AT3 domain to the SGNH domain in AT3-SGNH proteins (*Jones et al., 2021*). Furthermore, it is known that the catalytic triad of the SGNH domain of OafB from *Salmonella enterica* subsp. *enterica* serovar Typhimurium is required for O-acetylation of its acceptor molecule in the O-antigen of LPS (*Pearson et al., 2020*). Thus, in standalone AT3 proteins an equivalent, as yet unknown, partner protein may exist. If this were the case, it would suggest that the system may be similar to that of the PatA and PatB peptidoglycan acetyltransferase system. In this system it is hypothesised that the MBOAT (membrane-bound O-acyl-transferase) protein PatA transports the acetyl group across the membrane where it is transferred to the peptidoglycan substrate by PatB (an SGNH domain containing protein) (*Moynihan and Clarke, 2014*).

To further study the AT3-SGNH family of proteins, OafB from *S. enterica* subsp. *enterica* was used as a model system. The O-antigen of LPS consists of species-specific polysaccharide repeating units, which for serovar Typhimurium consists of repeating units of mannose, rhamnose, and galactose, with an abequose linked to the mannose residue, whereas serovar Typhi has a tyvalose linked to the mannose. OafB specifically O-acetylates the rhamnose moiety of this O-antigen repeating unit. As

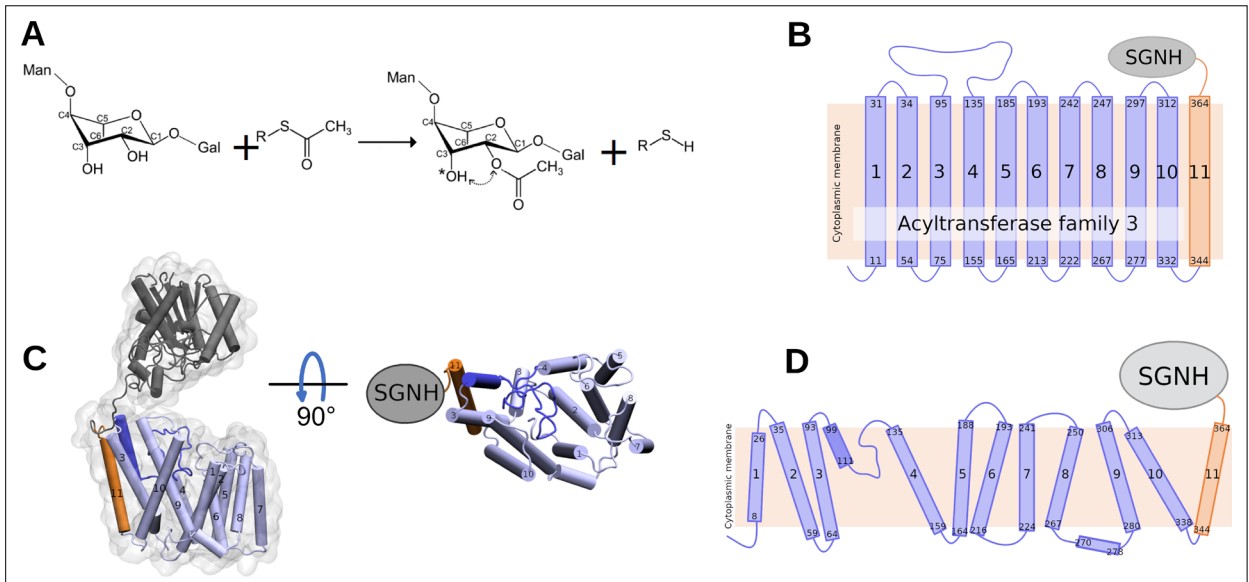

**Figure 1.** Reaction catalysed by OafB, and its predicted topology and structure. (**A**) Schematic of the acetylation reaction carried out by OafB. C2 of rhamnose in the repeating O-antigen unit (shown partially, see text) is O-acetylated, likely using acetyl-CoA as acetyl group donor. The acetyl group can migrate (dotted arrow) to the C3 (indicated by *). R = CoA; Man = mannose; Gal = Galactose (*Pearson et al., 2020*; *Micoli et al., 2014*) (**B**) TOPCONS topology prediction of OafB with N- and C-termini of each transmembrane helix (TMH) indicated. Consistent with the RaptorX structure, TOPCONS predicts short cytoplasmic loops and a long periplasmic loop between TMH 3 and 4. (**C**) RaptorX predicted structure of OafB with AT3 domain (TMH1-10) coloured light blue, loop between TMH3–4 coloured blue, TMH11 coloured orange, and the SGNH domain coloured grey (**D**) Topology schematic of OafB based on the RaptorX predicted structure. The RaptorX structure has 11 TMH, with a long periplasmic loop between TMH 3 and 4 consisting of a short helix followed by an unstructured region. TMH1–10 form the AT3 domain and are coloured blue, TMH11 forms the linking region between the AT3 and SGNH domains and is coloured orange.

The online version of this article includes the following figure supplement(s) for figure 1:

**Figure supplement 1.** OafB structure as predicted by RaptorX (yellow) and AlphaFold (lilac).

has been proposed for other AT3 proteins, cytoplasmic acetyl-CoA is the predicted acetyl group donor for this reaction (*Figure 1A*). Modification of the O-antigen has repeatedly been identified as important for virulence and persistence of bacteria (*Zou et al., 1999*; *Kahler et al., 2006*; *Fox et al., 2005*; *Kooistra et al., 2001*). Indeed, O-antigen acetylation of the abequose and rhamnose moiety of serovar Typhimurium, mediated by OafA and OafB, respectively, was found to vary significantly among clinical isolates (*Van Puyvelde et al., 2022*). Similarly, O-acetylation by OafB has been acknowledged in the development of vaccines against *Salmonella* ser. Paratyphi A (SPA), where acetyl groups were shown to be required to elicit bactericidal antibodies (*Konadu et al., 1996*; *Ravenscroft et al., 2015*). Furthermore, OafB-mediated acetylation in invasive non-typhoidal *Salmonella* alters susceptibility to bacteriophage (*Kintz et al., 2015*). OafB consists of the membrane-bound AT3 domain linked to an SGNH domain via an 11th TMH and periplasmic linking region (AT3-SGNH) (*Pearson et al., 2020*; *Figure 1B*). The catalytic triad of this periplasmic C-terminal SGNH domain is essential for O-antigen acetylation (*Pearson et al., 2020*). The structure of this SGNH domain and the periplasmic linking region was previously solved using x-ray crystallography (*Pearson et al., 2020*). The SGNH structure was found to be similar to that of other SGNH domains, and the periplasmic linking region formed a structured extension suggesting the AT3 and SGNH domains are likely to exist in close proximity and may even interact (*Pearson et al., 2020*).

Herein, we use RaptorX to determine a computationally derived structure for the transmembrane region, including the 10-TMH AT3 domain, of OafB. We show this novel structure to be stable under physiological conditions *via* molecular dynamics (MD) simulations. This model is integrated with the existing x-ray structure of the SGNH domain and published mutational data to allow additional structure-function analysis. Utilising both classical MD simulations and quantum mechanical calculations, we consider the acetyl donor and acceptor substrates to generate a refined functional model for this important family of membrane proteins.

## Results and discussion

### RaptorX model of OafB supports specific topology predictions and identifies new features

The structure of OafB was predicted by RaptorX (*Wang et al., 2016*; *Källberg et al., 2014*; *Ma et al., 2015*) using the protein sequence of a *Salmonella* rhamnose O-acetyltransferase (OafB). This method has been successfully validated with numerous proteins (*Sharma et al., 2021*; *Mariani et al., 2011*; *Lopes-Rodrigues et al., 2019*; *Bakar and Kaplan-türköz, 2017*; *Xu and Wang, 2019*). The structure of OafB consists of two key domains: the AT3 domain and SGNH domain (*Figure 1C*). The transmembrane region has 13 helices, of which 11 completely span the membrane (*Figure 1C and D*). The AT3 domain (residues 1–338) has 10 TMH and an additional 11th TMH (residues 339–376) fused to the periplasmic linking region (containing SGNH-ext, residues 377–421), which facilitates a periplasmic location of the fused SGNH domain (residues 422–640) (*Pearson et al., 2020*), as predicted from simple TOPCONS topological analysis (*Bernsel et al., 2009*; *Tsirigos et al., 2015*). The topological analysis also predicted a loop region between helices 3 and 4 (*Figure 1B*); in the RaptorX model, this region is clearly suggested to be a short re-entrant loop consisting of a short, well-structured helix (residues 99–111) followed by an ~24 residue-long unstructured region (*Figure 1C and D*). This means that there are no long hydrophilic loops in the structure on either side of the membrane (*Pearson et al., 2020*).

The structure of the SGNH domain and periplasmic linking region of OafB was previously solved using x-ray crystallography (*Pearson et al., 2020*). The RaptorX structure of the same region showed remarkable similarity, with a root-mean-square deviation (RMSD) of 2.8 Å. As seen in the crystal structure, the SGNH domain from OafB resembles a typical SGNH domain but with an additional helix and structured extension. Neither of these regions is seen in the structures of other SGNH domains in the RCSB PDB database; despite this, RaptorX predicts these regions with high accuracy. There are very few residues between the C-terminus of TMH11 (residue 367) and the N-terminus of the structured extension (residue 377), suggesting that the AT3 and SGNH domains are likely in close proximity and may interact (*Pearson et al., 2020*). However, in the RaptorX structure of OafB, the two domains show limited interaction and are not in close proximity (*Figure 1C*).

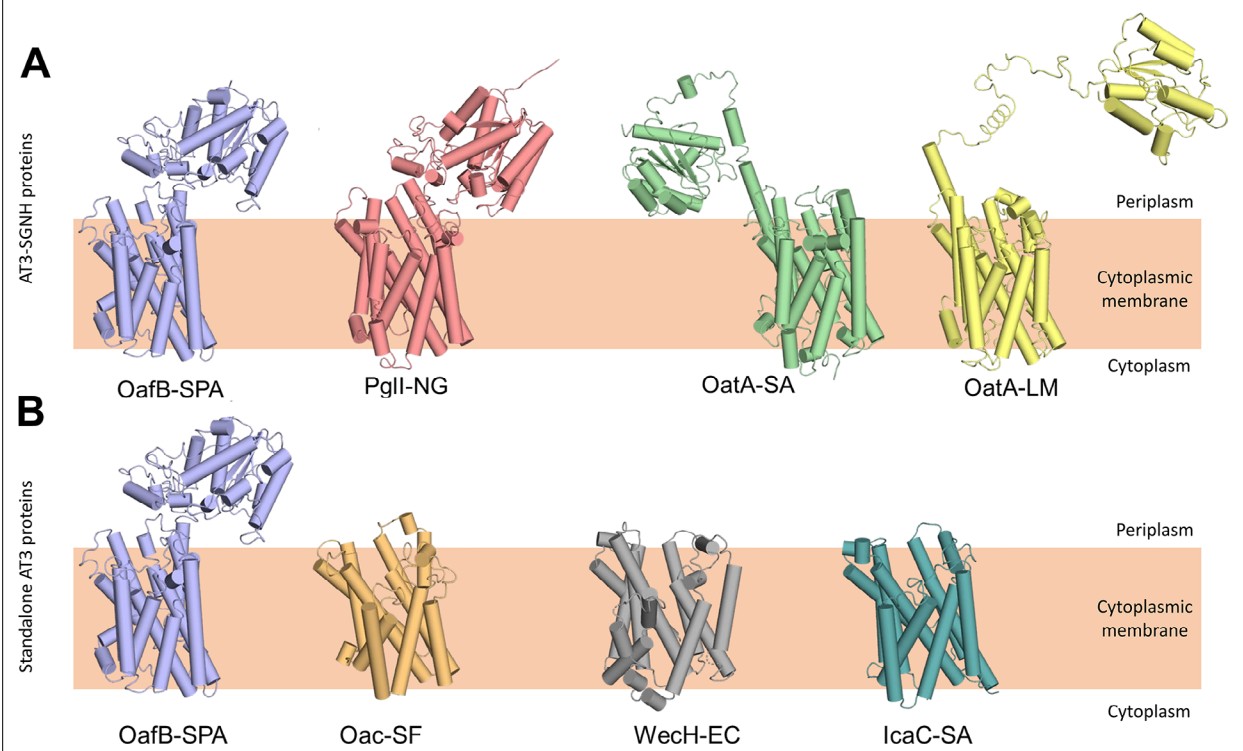

**Figure 2.** RaptorX predicted structures of AT3 domain containing proteins. (**A**) Fused AT3-SGNH proteins with OafB for comparison. Left to right: OafB from *Salmonella enterica* subsp. *enterica* ser. Paratyphi A (OafB); PglI from *Neisseria gonorrhoeae* (PglI-NG); OatA from *Staphylococcus aureus* (OatA-SA); OatA from *Listeria monocytogenes* (OatA-LM). (**B**) Standalone AT3 proteins with OafB for comparison. Left to right: OafB-SPA; Oac from *Shigella flexneri* (Oac-SF); WecH from *Escherichia coli* (WecH-EC); IcaC from *S. aureus* (IcaC-SA). The structure of the AT3 domain is largely conserved between these proteins.

Similar to RaptorX, AlphaFold (*Jumper et al., 2021*) predicted 11 TMHs, with the SGNH domain in the periplasm. The structures predicted for residues 1–338 (TMH1–10 of the transmembrane domain, the AT3 domain) showed 97% coverage (RMSD of 3.04 Å) and a template matching score of 0.85, indicating the same structure across the two models (*Figure 1—figure supplement 1*).

## Diverse bacterial AT3 domain-containing proteins share a common 10 TMH structure

Having built a model for our primary experimentally characterised protein, we expanded the analysis to include other important AT3 domain-containing proteins, including both additional AT3-SGNH fusion proteins and standalone AT3 proteins. The RaptorX generated structures of the AT3-SGNH proteins OatA from *S. aureus* (OatA-SA), OatA from *L. monocytogenes* (OatA-LM), and PglI from *Neisseria gonorrhoeae* (PglI-NG) all closely resembled the OafB (*Figure 2A*), with an RMSD of less than 3 Å for the AT3-domain over the 10 TMHs. Importantly, as they all contain a C-terminal SGNH domain that functions extra-cytoplasmically, they all contain an 11th TMH to enable correct localisation of the fused domain (*Figure 2A*; *Moynihan and Clarke, 2011*; *Pearson et al., 2020*; *Bonnet et al., 2017*; *Bernard et al., 2012*). In addition, the core of the SGNH domains is also similar (*Pearson et al., 2020*) with variation occurring in the length of the linking regions that connect the AT3 to the SGNH. While the PglI-NG protein closely resembles OafB with a short structured linker (*Pearson et al., 2022*), OatA proteins show a more extended, and potentially more mobile and flexible, structure, locating the SGNH domain further from the AT3 domain (*Figure 2A*).

Expanding the analysis to standalone AT3 proteins, the RaptorX outputs from Oac from *S. flexneri* (Oac-SF), WecH from *Escherichia coli* (WecH-EC), and IcaC from *S. aureus* (IcaC-SA) were analysed (*Figure 2B*). While the RMSD of these compared to the AT3-domain of OafB was more variable (RMSD of 4.5–6.7 Å when compared to residues 1–376 of OafB), the overall 10 TMH features were conserved and are consistent with the experimentally determined topology of OafB and Oac-SF (*Kintz et al.,*

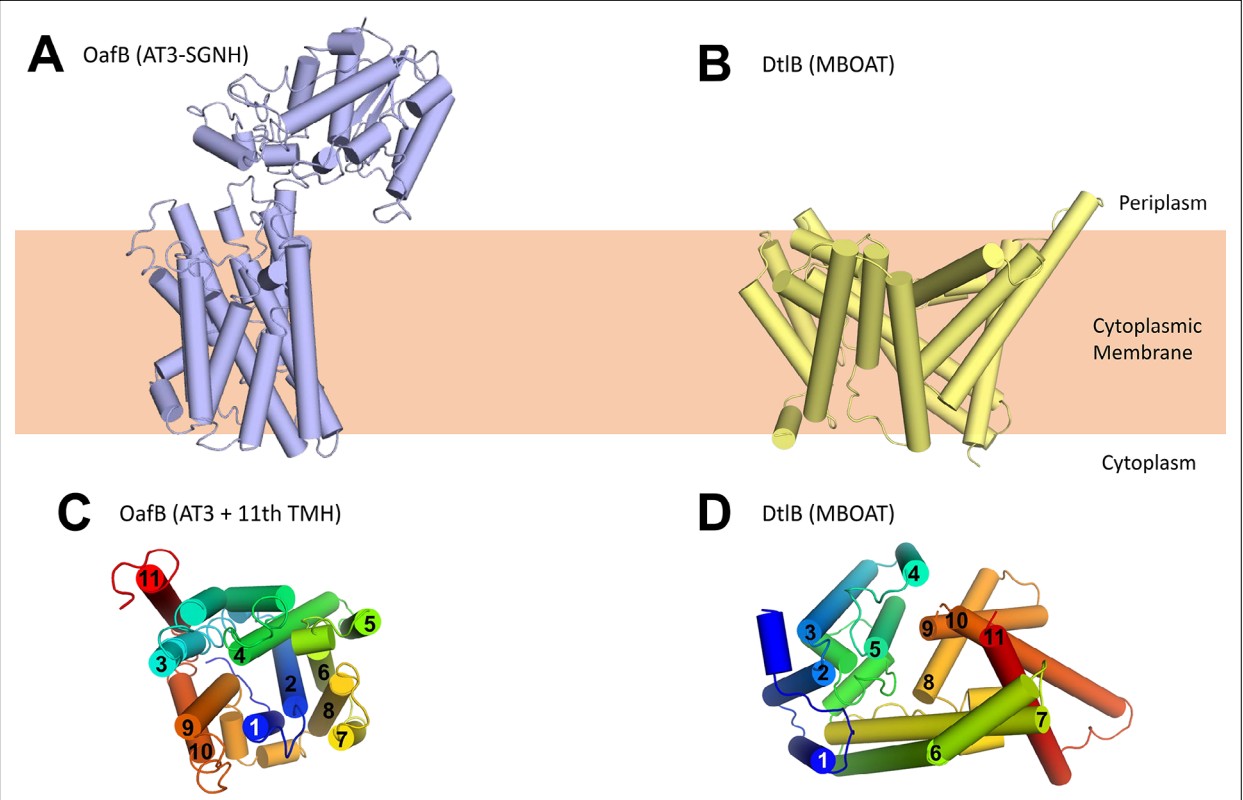

**Figure 3.** Structure of OafB, an AT3-SGNH protein (left, panels **A** and **C**), compared to DltB from *Streptococcus thermophilus*, an MBOAT (membrane-bound O-acyltransferase) protein (right, panels **B** and **D**). Both OafB and DltB have 11 TMH (**C, D**); however, the arrangements differ considerably. (**C**) OafB and (**D**) DltB viewed from periplasm, coloured blue to red, N- to C-termini with transmembrane helices numbered.

The online version of this article includes the following source data and figure supplement(s) for figure 3:

**Figure supplement 1.** Analysis of the RaptorX model for the transmembrane domain of OafB, embedded in a model *Escherichia coli* membrane and simulated under equilibrium conditions.

**Figure supplement 1—source data 1.** This file provides the source data for *Figure 3—figure supplement 1A*.

**Figure supplement 1—source data 2.** This file provides the source data for *Figure 3—figure supplement 1B*.

*2015*; *Thanweer et al., 2008*). However, a recent analysis of *S. aureus* OatA combining LacZ-PhoA fusion data with in silico predictions (*Meziane-Cherif et al., 2015*) led to an alternative model for the topology of the AT3 domain with nine TMH, a further re-entrant helix and a long cytoplasmic loop preceding the final TMH that lead to the SGNH domain in the periplasm. This topology is not seen in any of our models, including similar models generated in AlphaFold (data not shown), and these conflicts will be discussed later.

## The AT3 domain defines a novel and stable membrane protein fold

Using our new structural models for AT3 proteins, we assessed their similarity to known structures using the DALI server (*Holm, 2020*). The closest structural homologues consisted of small helical bundles that were fragments of larger proteins, and no significant matches to full length proteins were identified. Looking specifically at other membrane proteins with 10–11 TMH (3M73, 4J72, 4KJR, 1RH5), we could not see similarities with the AT3 structure (RMSDs between 12 and 24 Å), and the arrangement of TMH is very different. Finally we compared our AT3 structure to that of the other known family of membrane-bound acyltransferases, the MBOAT proteins (*Figure 3*). These also function in the acylation of complex extracytoplasmic carbohydrates (*Ma et al., 2018*), and previous analysis has shown an MBOAT protein to be functionally interchangeable with a standalone AT3 protein (*Kajimura et al., 2006*; *Moynihan and Clarke, 2010*; *Figure 3B*), again suggesting that these proteins have closely related functions. The structure of DltB (an MBOAT protein from *Streptococcus thermophilus*) (*Ma et al., 2018*) similarly contains 11 TMH; however, the helical arrangement is entirely

different (*Figure 3C and D*). Consistent with a core 10 TMH fold in the AT3-SGNH protein, the 11th TMH present sits on the outside of the bundle as would be expected for a non-essential feature of the AT3 domain (*Figure 3C*), in contrast to the 11th TMH in DltB (*Figure 3D*).

To investigate the biological plausibility of the proposed AT3 fold, we applied MD techniques to emulate the behaviour of the protein in a physiological environment and to extract meaningful data on the dynamics and structure-function relationships of the system. The RaptorX model of the transmembrane domain (TMH1–11 of OafB, residues 1–376) was embedded in a model *E. coli* inner membrane in 150 mM KCl solution. When subjected to MD simulations of 50 and 100 ns at 320 and 303 K, respectively, the protein model remained stable within the membrane. There was no significant unfolding observed; secondary structure analysis indicates that the alpha-helical segments of the transmembrane domain are maintained throughout, and the RMSD of the protein backbone remained below 0.5 nm at both temperatures (*Figure 3—figure supplement 1A, C*; *Figure 3—figure supplement 1—source data 1*). The root-mean-square fluctuation by residue, as expected, is the highest for the termini and for residues in unstructured loops, and lower for the TMHs (*Figure 3—figure supplement 1B*; *Figure 3—figure supplement 1—source data 2*). Together these data suggest that the computational approach described a novel, stable membrane protein fold for AT3 proteins.

## The AT3 domain from OafB has a cavity lined with essential residues forming a putative acetyl-CoA binding site

Having established that the AT3 domain constitutes a novel stable fold in the membrane, we then wished to use this structural model to try and understand the mechanism of this membrane-bound enzyme. A large cavity with a volume of ~3,620 $Å^3$ can be observed in the AT3 protein on the cytoplasmic side, which is enclosed by TMH 1, 2, 9, and the unstructured loop between TMH 5 and 6 (*Figure 4A*). The top end of this cavity is blocked by the novel re-entrant loop between THM 3 and 4, which in contrast to TOPCONS predictions (*Figure 1B*) is not in the periplasm but resides within the transmembrane domain (*Figure 4A*). Examination of the positioning of core amino acids known to be important and/or essential for the function of different AT3 proteins onto the predicted structure reveals that many of these in fact line this cytoplasmic cavity in the protein (*Figure 4C*). A channel lined with conserved essential residues is also observed in homologous AT3 proteins (*Figure 4—figure supplement 1*; *Figure 4—figure supplement 2*).

Two conserved motifs have been identified in AT3 domains: an absolutely conserved R-X$_{10}$-H motif located in TMH1 (Arg14 and His25 in OafB, *Figure 4C*; *Pearson et al., 2020*); and a highly conserved RXXR motif located on the cytoplasmic face of TMH3 (Arg71 and Arg74) (*Corvera et al., 1999*; *Kintz et al., 2015*; *Slauch et al., 1996*; *Thanweer and Verma, 2012*; *Figure 4—figure supplement 1*). The four residues found in these motifs have all been identified in experimental studies as being essential for function in OafB and other AT3 proteins (*Kintz et al., 2015*; *Spencer et al., 2017*; *Jones et al., 2021*; *Pearson et al., 2020*; *Calix et al., 2011*). Both arginine and histidine residues have previously been implicated in other acetyltransferase proteins as important for binding of acetyl-CoA (*Wu and Hersh, 1995*; *Ma et al., 2018*; *Jogl et al., 2004*). Perhaps significantly, all three Arg residues are in close proximity in positions approximately in the middle of the membrane, a usually unfavourable location, suggesting a key function in the mechanism of the enzyme. Further analysis of the cavity predicted in the model supports two other conserved regions as being involved, namely an FXXXXXL motif (Phe42-Leu48) in TMH2 and an SXXXY motif (Ser288 and Tyr292) in TMH9 (*Figure 4—figure supplement 1*).

In addition, Jones et al. performed site-directed mutagenesis on a number of residues in OatA from *S. aureus* (*Jones et al., 2021*). These identified, amongst others, three conserved tyrosine residues which are required for function (*Jones et al., 2021*). It is thought that Tyr206, located in the periplasm (equivalent residue in OafB Tyr194, *Figure 4C*), is involved in transfer of the acetyl group from the AT3 domain to the SGNH domain (*Jones et al., 2021*). Similarly, a catalytic triad consisting of this aforesaid tyrosine, plus a glutamic acid, and a histidine in the AT3 domain was proposed to remove the acetyl group from acetyl-CoA (*Jones et al., 2021*), allowing transport across the membrane (the three equivalent residues are highlighted in *Figure 4C* – Tyr194, His25, and Glu334). In the RaptorX structure of OafB, Tyr194 and His25 are in close proximity (C$_\alpha$ separated by 8 Å) and both sit on the periplasmic face of the membrane. However, Glu334 is located on the cytoplasmic side very distant from His25 and Tyr194, suggesting that it is not part of a catalytic triad.

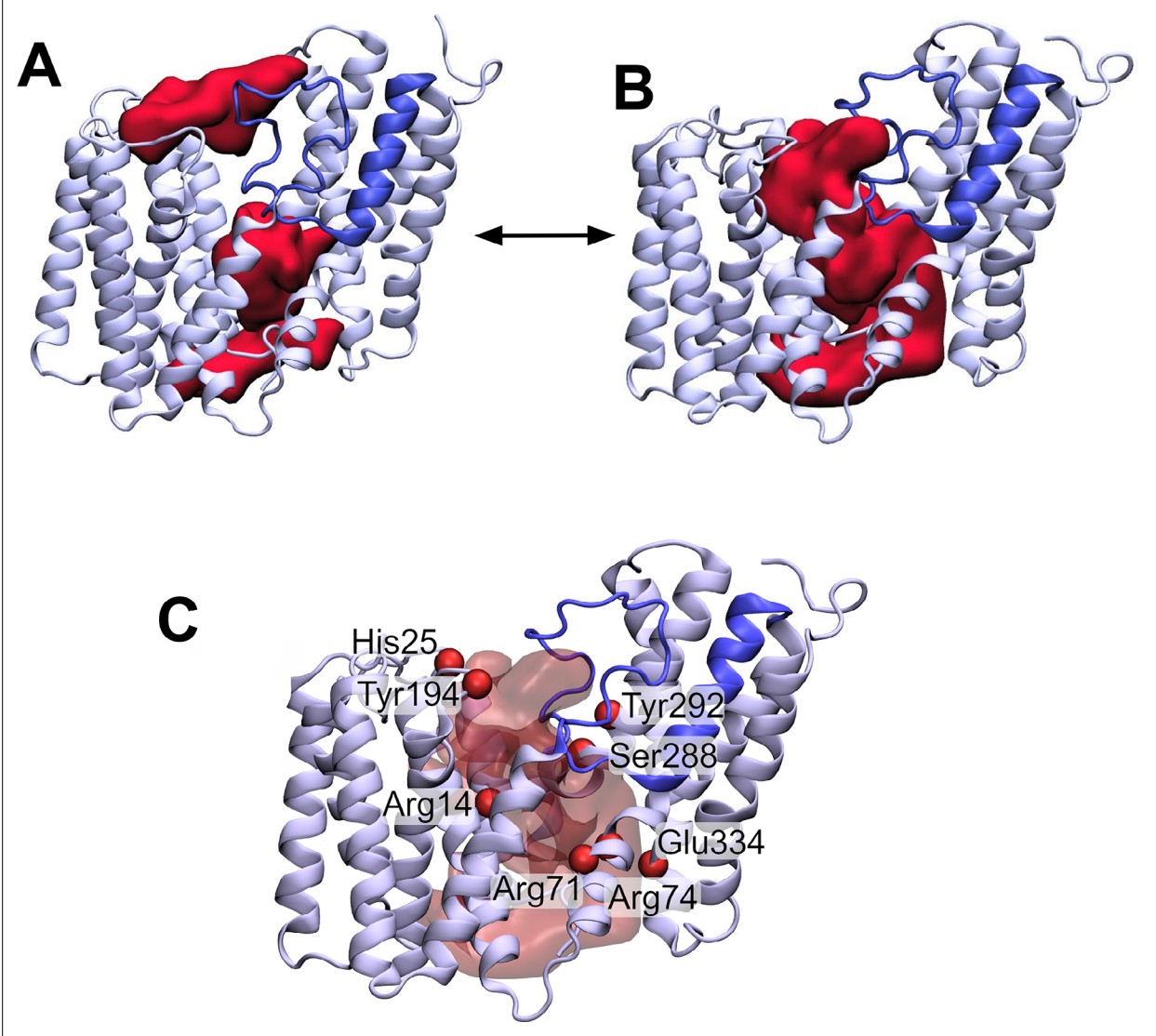

**Figure 4.** The loop between transmembrane helix (TMH) 3 and 4 controls the formation of a transmembrane channel lined with essential residues. (**A**) Initial equilibrated structure of the transmembrane domain (residues 1–376) of OafB. Residues 1–94 and 136–376 in pale blue, and the novel re-entrant loop residues 95–135 in blue. The two largest cavities identified by the ProPores2 server are shown as red surfaces. Initially, the loop between TMH 3 and 4 occludes the central pore in the AT3 domain. (**B**) Structure of the transmembrane domain after 50 ns equilibrium molecular dynamics at 320 K. The loop between TMH 3 and 4 is dynamic – this snapshot from the end of the simulation shows the loop to have moved away from the centre of the cavity to allow a channel to form. This channel could be occupied by the acetyl group donor. (**C**) Same snapshot as (**B**), but pore shown as a transparent red surface. Important conserved residues shown as red spheres: these residues line the pore in the centre of the AT3 domain.

The online version of this article includes the following figure supplement(s) for figure 4:

**Figure supplement 1.** Multiple sequence alignment of selected AT3 domain-containing proteins.

**Figure supplement 2.** AT3 domain-containing proteins, with conserved essential residues (identified via sequence alignment) mapped onto their respective RaptorX structure predictions (red spheres).

Together this mapping suggests two important sites in the protein, a mid-membrane region of positive charge created by three conserved Arg residues and a potential catalytic site on the periplasmic surface comprising at least His25 and Tyr194. In the RaptorX model, these two features of the protein are physically separated by the loop 3–4 region (*Figure 4A*). During equilibrium MD simulations, this loop is in fact dynamic (*Figure 4B*). Structures taken from later in the simulations indicate that the loop can move outwards (displacement of residues on the order of 3–4 Å) from the centre of the cavity towards TMH2, allowing the potential formation of a pore between the cytoplasmic and

periplasmic surfaces of the transmembrane domain, which could be a critical stage in the mechanism of the acyltransferase in the trans-membrane movement of acyl-groups (*Figure 4B*).

## The AT3 domain facilitates presentation of acyl-CoA molecules to the periplasm

Any model for an acyl-coa-dependent acylation of an extracytoplasmic acceptor sugar requires the transfer of the cytoplasmically located acyl group across the membrane. The inner cavity and dynamic conversion of this to a pore could provide the route for acyl-CoA molecules to enter the protein in a way to present the acyl group for use in the catalytic process (*Figure 5A*, left). OafB would use acetyl-CoA as the substrate, which is an extended molecule with a volume of ~630 $Å^3$. To test this hypothesis, one molecule of acetyl-CoA was pulled into this channel (entrance delimited by the N-terminus and TMH9 and 10) from the cytoplasmic entrance of the AT3 domain of a hybrid full-length OafB model (assembly discussed in the following section) using steered MD (SMD). It was found that this channel could indeed accommodate acetyl-CoA (*Figure 5A*, right) and, significantly, the thioester bond was positioned close to the essential His25 residue within the membrane. The conserved Arg14 was also positioned to help coordinate the 3'-phosphate of acetyl-CoA, as previously predicted (*Pearson et al., 2020*; *Figure 5A*).

The acetyl-CoA molecule was then allowed to equilibrate within the transmembrane domain using equilibrium MD. Three replicates of this system were generated and simulated with altered starting conformations for the acetyl-CoA within the protein. In the unrestrained final 20 ns of these simulations, acetyl-CoA was observed to hydrogen bond to several basic residues within the pore (*Figure 5B*). In addition to the Arg14 coordinating the phosphate of 3'-phosphate acetyl-CoA, we noted that the conserved Arg74 (of the RXXR motif of TMH3) along with Lys279 and Arg338 formed a small pocket capable of orienting such that they can all simultaneously hydrogen bond to the 3'-phosphate group (*Figure 5B*).

Furthermore, energy decomposition analysis (EDA) using ONETEP (*Skylaris et al., 2005*; *Prentice et al., 2020*) indicates strong, attractive interactions between the AT3 domain and acetyl-CoA. The transmembrane domain-acetyl-CoA complex was optimised, and a single point energy calculated. This EDA approach was employed to calculate the quantum mechanical interaction energy (in a vacuum) and decompose it into its contributing electronic components accounting for electrostatic, exchange, correlation, Pauli repulsion, polarisation, and charge transfer contributions. Adding the dispersion component, the overall interaction energy was calculated to be −1035.943 kcal mol$^{-1}$, of which the strongest interaction is electrostatics. This value was calculated in a vacuum and therefore does not translate to an absolute free energy of binding; rather here our intention is to highlight the energetically favourable nature of this interaction. Strong charge transfer interactions were identified between the protein and small molecule. The most prominent of these was between the 3'-phosphate of acetyl-CoA and nearby basic residues in the AT3 domain (Arg14, Arg74, Arg338, Lys279); these interactions are shown in *Figure 5C* and are in strong agreement with the above MD simulations.

We calculated the electrostatic potential of the transmembrane domain and acetyl-CoA and found them to be complementary: the transmembrane domain displayed positive potentials, while the acetyl-CoA displayed negative potentials, of similar magnitudes (*Figure 5D and E*). The greatest positive potentials are found towards the cytoplasmic side of the transmembrane domain (in agreement with the positive inside rule *Heijne, 1986*), and the greatest negative potentials are found at the phosphate groups of acetyl-CoA (which are positioned towards the cytoplasmic side of the AT3 domain). Together these data are strongly supportive of a model whereby acetyl-CoA from the cytoplasmic side is able to penetrate the enzyme to bring the acetyl group into range for use in catalytic transfer to the acceptor sugar on the periplasmic face of the membrane.

## Modelling suggests the two domains in OafB could function in a pedal bin mechanism

To learn more about the function of the model OafB proteins, which requires both the AT3 domain and its linked SGNH domain to function, we first constructed a hybrid structure that combined the RaptorX-predicted structure of the transmembrane domain (residues 1–376) with the x-ray crystal structure (380–640) of the SGNH-ext (380–421) and the SGNH domain (422–640). Residues 377–379 were added to the N-terminus of the SGNH domain using MODELLER 10.0 (*Webb and Sali, 2016*),

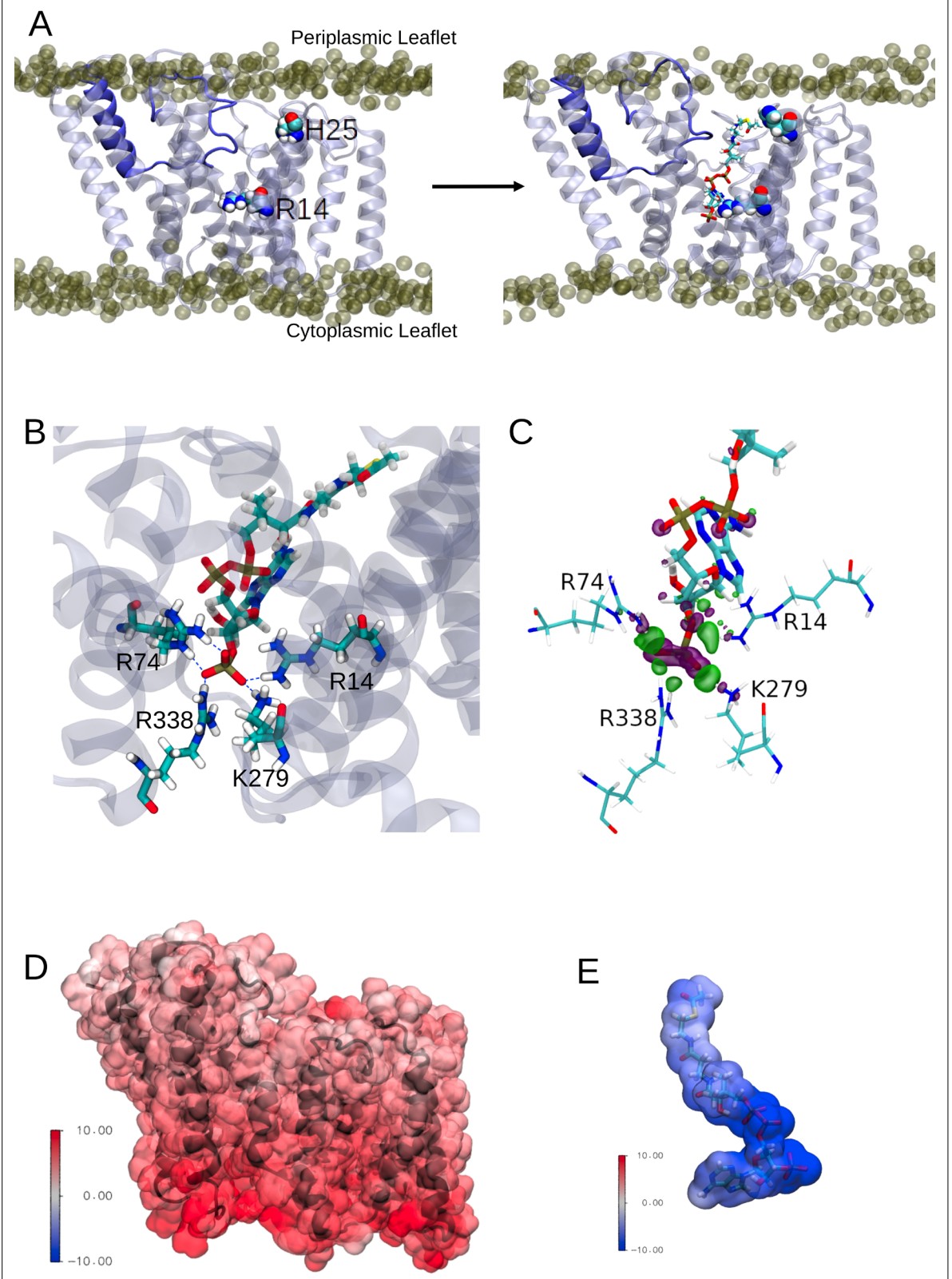

**Figure 5.** Interactions between the transmembrane domain of OafB and the putative acetyl donor molecule, acetyl coezyme-A. Note these structures are taken from simulations in which we use the full OafB protein, but only the transmembrane domain (residues 1–377) is shown here for clarity. (**A**) Left: Initial structure of the OafB transmembrane domain. Essential residues H25 and R14 shown as spheres, the loop between helices 3 and 4 is shown in dark blue. Phosphate headgroups of the phospholipids shown as tan spheres. The loop initially occludes the pore in the AT3 domain. Right: Structure

*Figure 5 continued on next page*

*Figure 5 continued*

from the end of the steered MD simulation in which acetyl coenzyme-A was pulled into the central channel within the AT3 domain. The loop between helices 3 and 4 has moved away from the centre of the pore towards transmembrane helix 2 (to the left) sufficiently to allow passage of acetyl-coenzyme A into the transmembrane domain. (**B**) A pocket of basic residues is observed in the AT3 domain, complementary to the 3'-phosphate of acetyl coenzyme-A. Several high occupancy hydrogen bonds are observed between the 3'-phosphate and transmembrane domain residues R14, R74, R338, and K279. (**C**) Charge transfer interactions between acetyl coenzyme-A and the transmembrane domain of OafB were identified via ONETEP Energy Decomposition Analysis. Loss of electron density is depicted as a purple surface and gain of electron density in a green surface. Significant charge transfer interactions were identified between the 3'-phosphate of acetyl coenzyme-A (losing electron density) and surrounding basic residues R14, R74, R338, and K279 (gaining electron density) (**D**) Electrostatic potential of the OafB transmembrane domain. Protein in black New Cartoon representation; calculated electrostatic potential overlayed in Surface representation. Scale from –10 V (blue) to +10 V (red). (**E**) Electrostatic potential of acetyl coenzyme-A. Molecule in Licorice representation; calculated electrostatic potential overlayed in Surface representation. Using the same scale as the AT3 domain, it is clear that acetyl coenzyme-A and the AT3 domain are complementary.

and a peptide bond was generated between residues 376 and 377 in ChimeraX (*Pettersen et al., 2021*) with varying C-N-C$_\alpha$-C dihedral angles (80, 100, 120, 140°) to generate proteins with differing relative conformations of the AT3 and SGNH domains (henceforth referred to as OafB$_{80}$, OafB$_{100}$, OafB$_{120}$, and OafB$_{140}$). To investigate the flexibility of the hybrid structure, we used equilibrium MD simulations (see Methods).

The x-ray crystal structure of the SGNH-ext region of OafB (PDB ID 6SE1, residues 380–421) (*Pearson et al., 2020*) shows it to be structured and interact substantially with the SGNH domain. The biological relevance of this close interaction is supported by the equilibrium MD simulations of OafB$_x$: the SGNH-ext remains structured and packed close to the SGNH domain throughout the 4×250 ns simulations, indicating this fold is stable under physiological conditions. Unfolding of the SGNH-ext to generate a longer flexible linker - such as that seen in the RaptorX-predicted structures of OatA-LM and OatA-SA - is not observed.

A short stretch of only around 10 residues (residues ~370–380) remains that could form a flexible linker between the transmembrane and periplasmic domains. Indeed in the MD simulations, the periplasmic domain exhibits a wide range of orientations relative to the transmembrane domain (*Appendix 1—figure 1*). Principal component analysis of the backbone atoms was used to extract the first two major motions (accounting 77% of the variance, *Table 1*) of the protein in the four equilibrium replicates. The first of these motions was a pedal bin-like action, where the periplasmic domain 'lid' opens and closes relative to the transmembrane domain, hinged at the flexible linker. The second motion was the rotation of the SGNH domain within the periplasmic space about the linker. The same first two major motions are observed when these four equilibrium simulations

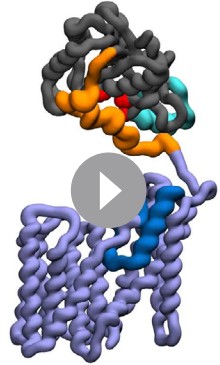

**Table 1.** Proportion of the variance accounted for by each eigenvector identified via principal component analysis of the equilibrium simulations.

| Eigenvector | % of variance |
| --- | --- |
| 1 | 53.2 |
| 2 | 23.8 |
| 3 | 11.1 |
| 4 | 3.8 |
| 5 | 2.9 |

**Video 1.** Motion described by the second principal component identified via principal component analysis. This represents a rotation of the SGNH domain within the periplasmic space about the flexible linker. Transmembrane domain coloured light blue; periplasmic loop between TMH 3 and 4 coloured dark blue; SGNH domain coloured grey with the periplasmic linking region in orange; additional helix in the SGNH domain in teal; and the SGNH catalytic triad in red.
https://elifesciences.org/articles/81547/figures#video1

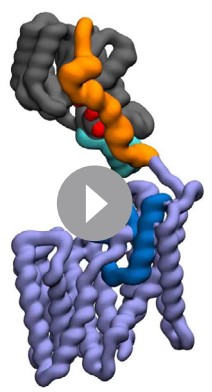

**Video 2.** Motion described by the first principal component identified via principal component analysis. This motion is 'pedal bin-like', with the periplasmic domain 'lid' opening and closing relative to the transmembrane domain, hinged at the flexible linker. Transmembrane domain coloured light blue; periplasmic loop between TMH 3 and 4 coloured dark blue; SGNH domain coloured grey with the periplasmic linking region in orange; additional helix in the SGNH domain in teal; and the SGNH catalytic triad in red.
https://elifesciences.org/articles/81547/figures#video2

are extended to 500 ns. The motions represented by PC1 and PC2 are shown as animations and as porcupine plots (*Appendix 1—figure 2*, *Video 1* and *Video 2*).

Despite these motions and the range of relative orientations, the SGNH and AT3 domains were not observed to spontaneously interact in the equilibrium simulations. This does not necessarily mean that the domains do not interact in vivo; given the short timescales of MD simulations and the vast conformational space available to the protein, sampling all possible conformations is an intractable problem. To direct computational efforts to the conformational space of interest, elastic network bonds were applied to pull the transmembrane and periplasmic domains together.

Elastic networks are a set of artificial harmonic 'bonds' added to a molecular model. These are commonly used in coarse-grained models to maintain the secondary structure of molecules (*Periole et al., 2009*); here, we use them to induce and then maintain a change in the tertiary structure of $OafB_x$ by pulling the periplasmic and transmembrane domains towards each other. The 'bonds' were added between residues likely to be proximal to one another as identified through coevolution analysis (see Methods). Around 75% of co-evolving residues have heavy atoms (carbon, nitrogen, oxygen, and sulfur) within 5 Å of one another (*Anishchenko et al., 2017*). The side chains of these residues were generally 3–5 Å in length; to avoid artificial re-orientation of the side chains but still bring the residues close enough to interact, elastic bonds of length 10 Å were added between the alpha carbons of the residue pairs. Applying elastic network bonds between the transmembrane and periplasmic domains of the $OafB_x$ proteins resulted in the rapid and irreversible 'closing' of the protein (the closed state), for the duration of the simulation time that the elastic network bonds were applied.

The elastic network bonds were then removed from these closed state structures, and over the subsequent 100 ns equilibrium MD simulations, all four replicates remained closed throughout. In this configuration, inter-domain interactions, sufficiently stable to maintain the closed state without the elastic network, are identified. Hydrogen bond analysis (distance and angle cut-offs of 3 Å and 20°, respectively) of the subsequent equilibrium simulations revealed a minimum of 35 unique hydrogen bonds between the periplasmic and transmembrane domains across each simulation. When averaged across all four simulations, the number of hydrogen bonds between the two domains at any given time was 3.16±1.67 (*Figure 6A*). Three of the higher occupancy hydrogen bonds found in all four replicates reside close to the linker region (*Table 2*, *Figure 6B*). Specifically, these are Lys132-Asp387 (all >51% occupancy), Thr386-Asp98 (>25% occupancy), and Ser129-Asp387 (>70% occupancy in three of the four simulations, 23% in the fourth). Additional hydrogen bonds were observed further from the linker in each system, on the other side of the catalytic triad: for example, Leu408-Glu189 (34% occupancy) and Ser474-Glu189 (21%) are observed in the $OafB_{100}$ system (*Figure 6C*) but do not form simultaneously (*Figure 6—figure supplement 1*). Taken together, these findings from equilibrium MD simulations identify interactions between the residues in the AT3 membrane domain and the periplasmic domains, supporting the hypothesis that the two domains may cooperate.

These fully closed structures were submitted to the ProPores2 webserver (*Hollander et al., 2021*) to identify and parameterise cavities within the protein. The closing of the pedal bin converts the outer cavity from the transmembrane domain into a larger enclosed cavity at the interface of the AT3 and

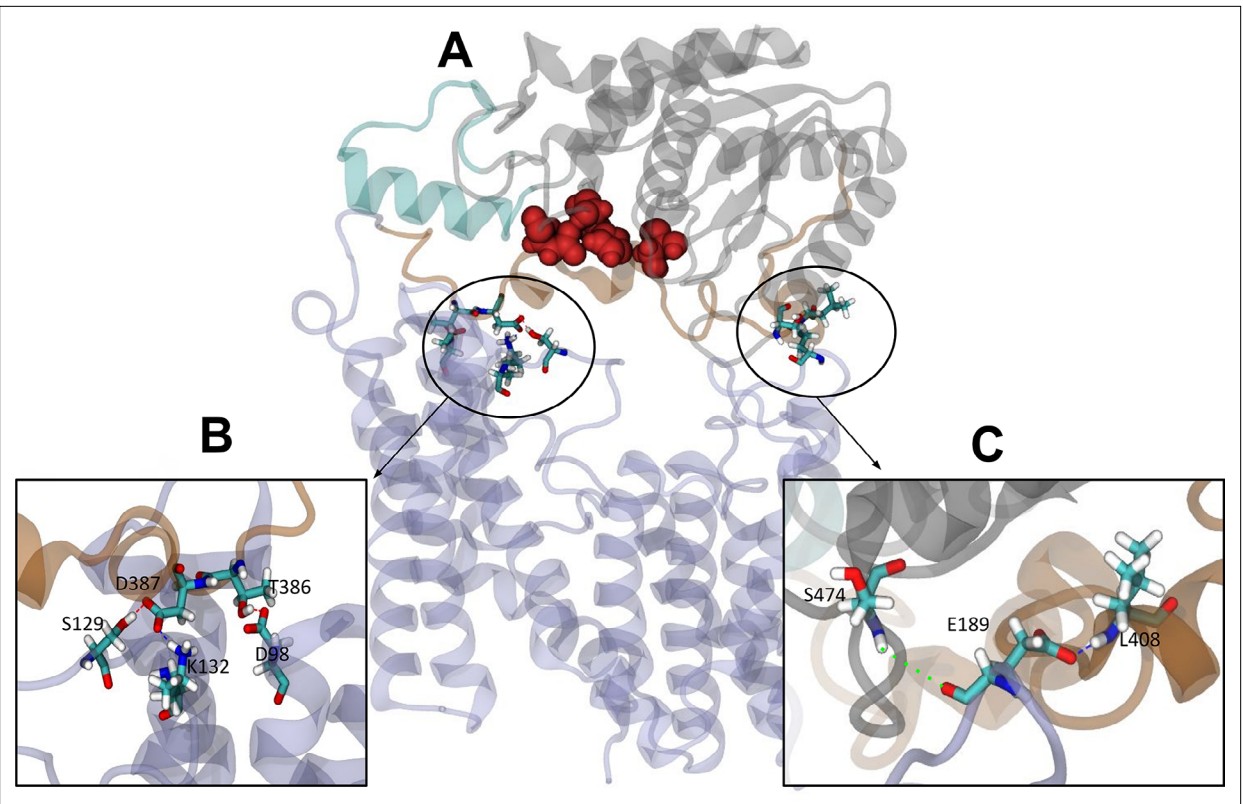

**Figure 6.** Interactions between the transmembrane and periplasmic domains in the closed state of OafB. (**A**) Closed OafB structure in New Cartoon representation (TM AT3 domain in lilac, SGNH-ext in orange, extra helix in SGNH domain (α8) in teal, and remainder of SGNH domain in grey). Catalytic triad (D618, H621, and S430) as red spheres. Residues identified in high-occupancy or important hydrogen bonding interactions between the SGNH and AT3 domains in Licorice representation. (**B**) Hydrogen bonding observed between SGNH-ext and AT3 domains in all equilibrium simulations of the closed OafB structure. High-occupancy hydrogen bonding between side chain: carboxylate of E387 and hydroxyl of S129; amine of K132; carboxylate of E98 and hydroxyl of T386. (**C**) Hydrogen bonding between the SGNH and AT3 domains observed in the OafB$_{100}$ simulations. E189 can hydrogen bond to S474 via its backbone carbonyl (green dashed line), or to L408 via its carboxylate side chain (blue dashed line) but cannot form these interactions simultaneously. Time series of the separation of these residues shown in *Figure 6—figure supplement 1*.

The online version of this article includes the following figure supplement(s) for figure 6:

**Figure supplement 1.** Separation between the carboxylate of E189 and backbone NH of L408 (teal), and backbone O of E189 and backbone NH of S474 (indigo).

SNGH domains (*Figure 7A*) This pore has a volume of ~4780 Å$^3$ and includes both the SGNH catalytic triad comprising residues Asp618, His621, and Ser430 (*Figure 7B*) and at the membrane surface the proposed catalytic residues of the AT3 domain, namely His25 and Tyr194. This raises the possibility that the transfer of the acyl-group to the acceptor molecule could occur in this region of the protein.

**Table 2.** High occupancy hydrogen bonds identified during the equilibrium simulations of the closed state of OafB.

| Hydrogen bond | Simulation(s) | % Occupancy |
| --- | --- | --- |
| Lys132-Asp387 | All replicates | >51 |
| Thr386-Asp98 | All replicates | >25 |
| Ser129-Asp387 | Three of four replicates | >70 (23 in other sim) |
| Leu408-Glu189 | OafB$_{100}$ | 34 |
| Ser474-Glu189 | OafB$_{100}$ | 21 |

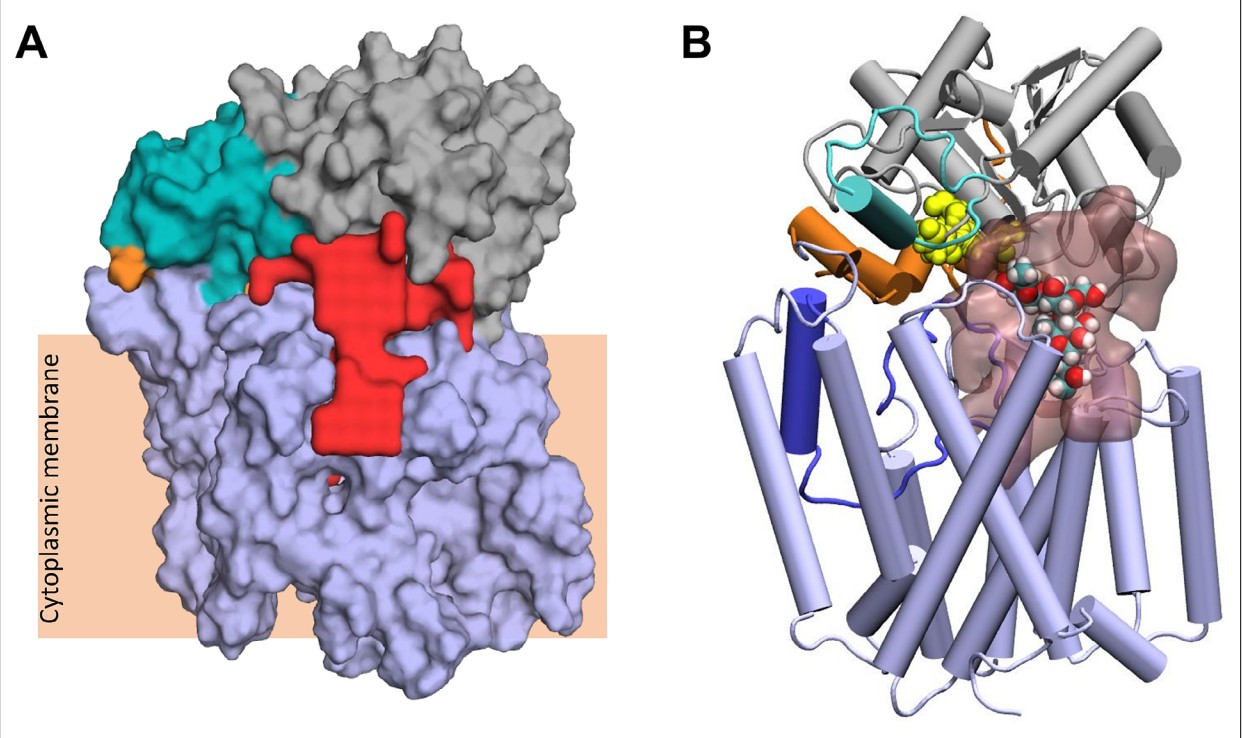

**Figure 7.** Largest cavity in the closed state of OafB. (**A**) Largest cavity identified by the ProPores2 Webserver in the closed OafB structure shown as a red surface. The cavity extends down into the transmembrane domain. (**B**) Closed OafB in Cartoon representation: transmembrane domain coloured light blue, periplasmic loop between TMH 3 and 4 coloured dark blue, SGNH domain coloured grey with the periplasmic linking region in orange, and additional helix in the SGNH domain in teal. The SGNH catalytic triad (Ser430, Asp618, and His621) is shown as yellow spheres. Largest pore identified by the ProPores2 Webserver shown as a transparent pink surface. A single O-antigen unit is shown inside the pore in van der Waals representation, coloured by atom name (cyan for carbon; red for oxygen; white for hydrogen). The single O-antigen unit can be comfortably accommodated in this space in several orientations.

## Fully closed structure exhibits cavity able to accommodate the LPS O-antigen

Finally, we considered the potential mechanisms by which the acceptor molecule(s) would interact with open and closed 'pedal bin' model of OafB, working with two alternative hypotheses that the acceptor is presented either as a single or multiple O-antigen repeat unit(s) anchored to a lipid carrier or later after transfer to the lipid A core molecule.

The closed form of the pedal bin model presents a possible enclosed catalytic site for OafB, which is known to acetylate the rhamnose residue in the O-antigen repeating unit of the LPS (*Kintz et al., 2017*). This cavity resides directly below the catalytic triad (Ser430, His621, and Asp618) and extends down into the AT3 domain (*Figure 7A*). The measured volume of ~4780 Å³ is an order of magnitude greater than the volume of the LPS O-antigen unit (volume of ~502 Å³); hence, one or more units could be manually placed within the cavity (*Figure 7B*). Importantly, the enclosed cavity on the periplasmic surface of the membrane does extend into the membrane, the lower portion residing below the plane of the upper leaflet headgroups, and hence, is exposed to the membrane core (*Figure 7A*), consistent with a ligand delivered on a lipid anchor.

*S. enterica* serovars Typhimurium and Paratyphi use the prevalent Wzx/Wzy-dependent pathway for O-antigen assembly, in which a single O-antigen unit is assembled in the cytoplasm on undecaprenyl phosphate (Und-PP) in the membrane, which is then flipped placing the O-antigen unit in the periplasm. Lipid-linked O-antigen units are assembled to the final length in the periplasm before ligation to lipid A-core. Subsequently, the completed LPS molecule is transported to the outer membrane (*Whitney and Howell, 2013*). Thus, the question remains, does OafB acetylate the rhamnose moiety of a single O-antigen unit bound to Und-PP or after O-antigen polymerisation? If the latter, does

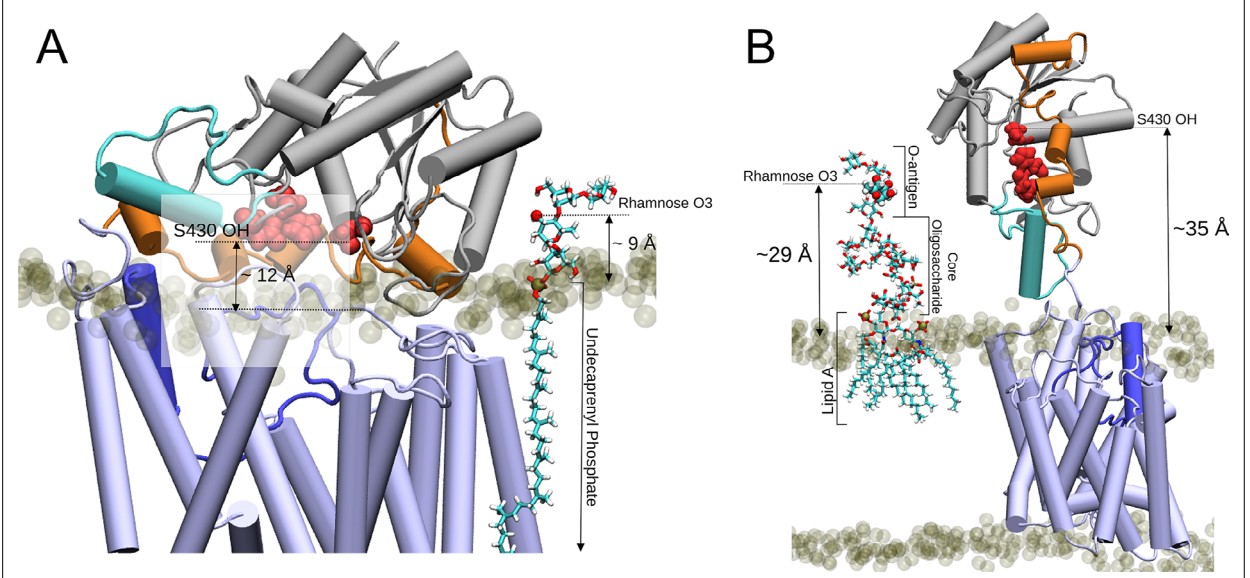

**Figure 8.** Two proposed models for the O-antigen rhamnose moiety acetylation by OafB. In both panels, OafB is shown in a cartoon representation with the AT3 domain coloured light blue, periplasmic loop between TMH 3 and 4 coloured blue, SGNH domain coloured grey, the periplasmic linking region in orange, and additional helix in the SGNH domain in teal. Catalytic triad of the SGNH domain (Ser430, Asp618, and His621) shown as red spheres. 1-palmitoyl-2-oleoyl-sn-glycero-3-phosphoethanolamine and 1-palmitoyl-2-oleoyl-sn-glycero-3-phosphoglycerol phosphate headgroups shown as transparent tan spheres. (**A**) Undecaprenyl phosphate carrier lipid with a single O-antigen unit in Licorice representation. O3 of the rhamnose moiety (marked by a red sphere) has a vertical separation from the periplasmic leaflet surface of ~9 Å. The OH of S430 has a vertical separation from the bulk membrane of ~12 Å. (**B**) OafB-lipopolysaccharide (LPS) system after equilibration steps - one LPS molecule with a single O-antigen unit shown in Licorice representation, and its rhamnose moiety shown in van der Waals representation. OafB remains in an open conformation. After relaxation, the oligosaccharide of LPS has kinked, with the distance between the lipid headgroups of the periplasmic leaflet surface and the acetylation site reduced to ~29 Å (from ~42 Å, as shown in *Figure 8—figure supplement 1*). This is comparable to the ~35 Å separation between the OH of S430 and the periplasmic leaflet surface.

The online version of this article includes the following figure supplement(s) for figure 8:

**Figure supplement 1.** One molecule of lipopolysaccharide (LPS) with a single O-antigen unit from *Salmonella enterica* subsp. *enterica* ser.

acetylation occur with the polymerised OAg unit(s) attached to the Und-PP or after transfer to the lipidA-core molecule?

To contextualise the OafB models with biologically relevant molecules, a system containing one molecule of the lipid carrier Und-PP attached to a single O-antigen unit was placed next to the OafB model (*Figure 8A*). A distance of ~9 Å separates O3 of the rhamnose from the periplasmic surface of the membrane, which is similar to the distance of the SGNH active site (Ser430) from the membrane (which was ~12 Å calculated with $OafB_{100}$ after 100 ns with elastic network bonds applied), meaning it is possible that the substrate diffuses laterally into the cavity, either with the pedal bin open or closed.

The analysis above suggests that acetylation could occur using single O-antigen units, before O-antigen polymerisation. Alternatively, acetylation occurs during or after polymerisation on the Und-PP carrier before transfer to the lipid A core (not shown). The distance of multimers of 9 Å (single O-antigen unit) stretching away from the membrane could possibly be accommodated by the open state of the pedal bin with the SGNH domain extending into the periplasm.

To investigate the final model of acetylation after assembly onto the lipid A core, we built a model with LPS (with one O-antigen unit) and OafB-SPA embedded in the inner membrane, to assess the distances defined between the periplasmic leaflet surface and various groups within the system relevant to the acetylation (*Figure 8B*). One molecule of *Salmonella* LPS was generated in CHARMM-GUI's LPS Modeler and inserted into the periplasmic leaflet of the $OafB_{100}$ system (replacing two 1-palmitoyl-2-oleoyl-sn-glycero-3-phosphoethanolamine [POPE] and two 1-palmitoyl-2-oleoyl-sn-glycero-3-phosphoglycerol [POPG] molecules). Initially the modelled LPS adopts a quasi-linear conformation; the z-separation between the rhamnose moiety and the periplasmic membrane surface was found to be ~42 Å, as shown in *Figure 8—figure supplement 1*. During

minimisation and equilibration, the oligosaccharide was allowed to relax; this resulted in a kinked structure, where the distance between the acetylation site and the periplasmic leaflet surface was reduced to around 29 Å (*Figure 8B*). By comparison, after equilibration, the z-separation of the hydroxyl of Ser430 in the catalytic triad of OafB$_{100}$ and the periplasmic leaflet is ~35 Å; hence, it is not inconceivable that the open pedal bin could function to deliver acetyl groups at a distance from the membrane. However, we note that in reality the full length O-antigen will be attached to the lipidA core, which would require significant, entropically unfavourable conformational rearrangement of the O-antigen for OafB to catalyse significant overall levels of O-acetylation across the whole polymer.

SMD was used to pull the rhamnose moiety of LPS and Ser 430 of OafB towards one another, and subsequent restrained MD simulations held the protein and LPS close to one another to assess possible interactions. Many hydrogen bonds were observed between the O-antigen and the periplasmic domain of OafB over the two replicates. Several of these were with residues from the SGNH-ext (galactose-Asp387; galactose-Tyr389; rhamnose-Tyr394). Tyr389 and Tyr394 are both residues which were previously identified by FTMap in silico docking analysis as involved in binding to rhamnose sugars. Hydrogen bonding was also observed between the 3' and 2' hydroxy groups of the rhamnose moiety and Ser430. Even when all position restraints were removed in the MD simulations, this interaction persisted for a further 10 ns.

Together these analyses indicate that O-acetylation of the O-antigen may occur either when a single O-antigen unit attached to a lipid carrier is presented in the periplasm, or when the O-antigen is polymerized, either on the lipid carrier, or when attached to the lipid A core.

## Ideas and speculation

The realisation that diverse bacteria use AT3 proteins in a plethora of different biological processes for the modification of extracytoplasmically localised glycans, directed our efforts into describing the fundamental mechanism of these proteins (*Pearson et al., 2022*). While the role of AT3 proteins in these diverse processes is often implicated from genetics only, the OafB protein has been studied in much more detail, along with other selected AT3 proteins including the OatA protein from *S. aureus* (*Kintz et al., 2017*; *Jones et al., 2021*; *Pearson et al., 2020*). These systems are examples of AT3 proteins working with SGNH domains to acylate LPS and peptidoglycan, respectively, which are key components of bacterial envelopes.

Integration of experimental and computational data has provided detailed insight into the structure and potential mechanism of action of AT3-SGNH proteins. Our structural model, combined with MD simulations, suggests first that the transmembrane domain of OafB has a novel fold. Importantly this is distinct from the MBOAT family of membrane-bound acyltransferases, suggesting that nature has evolved two independent routes to solve this problem. The discovery of a large cytoplasmic cavity, which can open dynamically to accommodate an acyl-CoA molecule that spans the membrane, provides a key mechanistic breakthrough in our understanding of the protein function, which in OafB is mediated by a loop region which would need to block the pore in the absence of donor/acceptor to prevent leakage of protons.

The putative catalytic residues His25 and Tyr194 are close enough to the acetyl group (~5 Å) to conceivably be involved in catalysis, although the MD simulations cannot capture bond making or breaking (nor proton transfer). The loop between TMH 5 and 6, in which Tyr194 resides, shows significant mobility and the side chain of Tyr194 is observed to point inwards, towards the centre of the pore (and therefore towards acetyl-CoA). Given the conformational mobility of the 4'-phosphopantetheine segment of the coenzyme, and of the loop between TMH 5 and 6, it is not unreasonable to suggest that the thioester bond is accessible to both His25 and Tyr194 to aid the acetyl transfer in a mechanism such as that suggested by *Jones et al., 2021*. Addressing the role of Glu334, which *Meziane-Cherif et al., 2015* proposed was the third residue in the catalytic site (Glu357 in OatA), our model places this distantly from the His25 and Tyr194, but yet in OafB we also know this residue is important for function (*Pearson et al., 2020*). The residue sits close to the cytoplasmic side and the internal cavity, and perhaps, has a role in the initial recognition of the acyl-CoA. Other conserved residues, including Phe42, Tyr124, Trp138, and Glu143 in OafB, which were identified by mutagenesis as being important for both the OafB-related protein OafA and OatA, cluster on the extracytoplasmic surface and could have related roles in acyl-CoA binding, catalysis, or acceptor recognition.

While the function of the AT3 to bind and deliver acyl-groups to a periplasmic catalytic site is well supported by our model and must be a conserved feature of all AT3 proteins, the further discrete role of the SGNH domain is still not fully understood, although it is essential for transferase function in both OafB and OatA (*Jones et al., 2021*; *Pearson et al., 2020*). The linker region, while short and structured in OafB suggests that the domain can function in a 'pedal bin' like model, opening and closing to define a large cavity on the extracytoplasmic surface, which we demonstrate could then capture a lipid-anchored O-antigen repeat structure. For OatA and other related proteins, this linker can be much longer (*Figure 2*) and whether this then allows the pedal bin lid to close and accept the acyl-group and then open and extend away from the membrane to reach the acceptor substrate needs further investigation, but the MD simulations suggest it is possible. However, for the OafB system, we favour the model of lid opening and closing to help trap the substrate in a membrane-surface-located activity site, similar to the function of the Sus protein in capturing ligand in the outer membrane of *Bacteroidetes* bacteria (*Glenwright et al., 2017*). For LPS O-acetylation by OafB and similar proteins we favour the model where the substrate is the single O-antigen repeat bound by the Und-PP lipid anchor before later polymerisation. Given that overall levels of OafB-dependent O-acetylation vary between 40 and 70% (*Kintz et al., 2017*; *Micoli et al., 2014*), the stochastic interaction of OafB with the Und-PP anchored O-antigen repeat appears a more parsimonious mechanism than relying on the enzyme trying to reach rhamnose sugars in a fully assembled O-antigen.

Beyond LPS and peptidoglycan, AT3 domain-containing proteins acylate a diverse range of complex carbohydrates in bacteria (*Pearson et al., 2022*). Each acceptor substrate may present its own unique challenges for the AT3 domain. In both the LPS and peptidoglycan examples, additional SGNH domains are required for function; however, AT3 domains can function as standalone proteins. One example of this is IcaC, which is involved in the O-succinylation of poly N-acetylglucosamine (PNAG) biofilm component of *Staphylococci*. Here, the acceptor is not delivered on a lipid anchor but rather through a continuous synthesis and extrusion mechanism catalysed by IcaAD, a 'synthase' mechanism (*Whitney and Howell, 2013*). Perhaps a close physical association of IcaC with IcaAD removes the need for the additional SGNH domain and allows direct transfer of the acyl group onto the emerging PNAG polymer giving stochastic levels of O-succinylation of about 40% (*Sadovskaya et al., 2005*; *Joyce et al., 2003*). Hence, the differences between AT3 and AT3 with additional domains such as SGNH, perhaps could relate to the route of presentation of the acceptor molecule within the context of different biosynthetic pathways for extracytoplasmic glycans.

## Conclusions

Here we provide strong evidence that the integral membrane AT3 domain (PF01757), mediating acylation of a diverse range of complex carbohydrates, has a novel fold. The stability of this structure was confirmed using MD simulations, which also enabled insights into the mechanism of action of AT3 domains. Importantly, for the first time, our model provides a solution to the problem of acetylation occurring in the periplasm using the cytoplasmic acetyl donor acetyl-CoA. We show a membrane-spanning pore occurs transiently in the AT3 domain which can accommodate acetyl-CoA, presenting the acyl group to the periplasmic side. This part of our model is supported by quantum-level calculations to probe the hypothesised protein-acetyl-CoA interactions, and the presence of key, conserved and essential residues that line the pore. The second cavity at the periplasmic side in OafB is able to accommodate the acceptor substrate O-antigen. Together, our data give important new insights into the family of AT3 domain containing proteins. Our well-supported model can support and drive targeted research to gain a full understanding of the mechanism of action of this family or membrane-bound acyltransferase proteins that mediate acylation of complex carbohydrates across the domains of life.

## Methods

### Co-evolution and protein structure predictions

OafB protein sequence (Uniprot accession A0A0H2WM30) was submitted to either the RaptorX contact prediction server (*Wang et al., 2016*; *Källberg et al., 2014*; *Ma et al., 2015*) or AlphaFold server (*Jumper et al., 2021*), and analysis was run with default parameters. Predicted structures using

RCSB PDB Pairwise Structure Alignment webserver, employing the flexible jFATCAT alignment algorithm (*RCSB PDB, 2021*).

## Molecular dynamics simulations

All simulations used the GROMACS simulation package (version 2020.3) (*Van Der Spoel et al., 2005*). Results were analysed using GROMACS tools and in-house python scripts utilising MDAnalysis (*Michaud-Agrawal et al., 2011*; *Gowers et al., 2016*; *Harris et al., 2020*; *McKinney, 2011*). Molecular graphics were generated using VMD 1.9.4 (*Humphrey et al., 1996*).

## Atomistic simulations

Atomistic simulations used the CHARMM36m forcefield (*Huang and MacKerell, 2013*) with the TIP3P water model (*Jorgensen et al., 1983*). A cut-off of 1.2 nm was applied to Lennard-Jones interactions and short-range electrostatics using the potential shift Verlet scheme. Long-range electrostatics were treated using the particle mesh-Ewald method (*Essmann et al., 1995*). Atoms were constrained using the LINCS algorithm to allow the use of a 2-fs timestep (*Hess et al., 1997*). For production simulations, temperatures were maintained using the Nosé-Hoover thermostat (*Nosé, 1984*; *Hoover, 1985*) (1.0 ps coupling constant), and pressure was maintained at 1 bar using the Parrinello-Rahman semi-isotropic barostat (*Parrinello and Rahman, 1981*) (5.0 ps coupling constant).

## Transmembrane domain model

A model of the transmembrane domain (residues 1–376, i.e. the 10 TMH of the AT3 domain and the 11th TMH) was generated using the RaptorX webserver (*Wang et al., 2016*; *Källberg et al., 2014*). Using the CHARMM-GUI membrane builder module (*Lee et al., 2019*), this structure was embedded within a model *E. coli* inner membrane: a symmetric bilayer of 18:1:1 POPE, POPG, and 1',3'-bis[1-palmitoyl-2-oleoyl-sn-glycero-3-phospho]glycerol (cardiolipin). The bilayer system was subsequently solvated in 0.15 M KCl. The system was energy minimised in 5000 steps using the steepest descent method (*Goldstein, 1965*). The subsequent structure was equilibrated in six phases in which the protein and lipid head groups were subjected to position restraints with varying force constants. The full equilibration protocol is detailed below. Two short, independent equilibrium MD simulations were carried out on the equilibrated system: one simulation of 100 ns at 303.15 K, and one of 50 ns at 320 K.

## Equilibration protocol

Each system was equilibrated using two NVT (canonical ensemble) phases followed by four NPT (isothermic-isobaric ensemble) phases. Details for the position restraints used are in *Table 3*. NVT phases used a timestep of 1 fs and each lasted 0.125 ns, and the NPT stages used a timestep of 2 fs and each lasted 0.5 ns. The velocity-rescaling thermostat (*Bussi et al., 2007*) was applied at all stages to bring the system to 303.15 K with a coupling constant of 1.0 ps. Semi-isotropic Berendsen pressure coupling (*Berendsen et al., 1984*) was applied to the NPT phases to equilibrate with a pressure bath of 1 bar ($\tau_P$ = 5.0 ps, compressibility of $4.5 \times 10^{-5}$ bar$^{-1}$).

**Table 3.** Position restraints used in the equilibration steps for all atomistic systems.

| Equilibration phase | Position restraint / kJ mol$^{-1}$ nm$^{-2}$ | | | |
| --- | --- | --- | --- | --- |
| | Protein backbone | Protein sidechains | Lipid head groups | Dihedrals |
| NVT1 | 4000 | 2000 | 1000 | 1000 |
| NVT2 | 2000 | 1000 | 400 | 400 |
| NPT1 | 1000 | 500 | 400 | 200 |
| NPT2 | 500 | 200 | 200 | 200 |
| NPT3 | 200 | 50 | 40 | 100 |
| NPT4 | 50 | - | - | - |

**Table 4.** Residues identified through co-evolution as likely to be proximal to one another.

| Residue 1 (TM domain) | Residue 2 (periplasmic domain) | % Probability |
|---|---|---|
| 94 | 545 | 56.1 |
| 95 | 546 | 80.5 |
| 96 | 546 | 57.2 |
| 97 | 546 | 70.5 |
| 98 | 546 | 58.4 |
| 123 | 458 | 53.3 |
| 125 | 403 | 59.2 |
| 126 | 404 | 55.9 |
| 194 | 627 | 50.9 |
| 242 | 503 | 51.3 |

## Full OafB model

The RaptorX transmembrane domain model and periplasmic domain crystal structure (PDB ID: 6SE1) were used to build a full OafB model. Missing linker residues (residues 377–379) were modelled into the N-terminus of the SGNH domain using MODELLER 10.0 (*Webb and Sali, 2016*). A peptide bond was generated between the N-terminus of the resulting periplasmic domain (K377) and the C-terminus of the trans-membrane domain (N376) using UCSF ChimeraX (*Pettersen et al., 2021*). Four structures were generated, varying the C-N-$C_\alpha$-C dihedral angle; values of 80, 100, 120, and 140° were used to give structures with different relative orientations of the two domains. The $C_\alpha$-C-N-$C_\alpha$ dihedral was maintained at 170°. These proteins will henceforth be referred to as $OafB_x$, where x is the C-N-$C_\alpha$-C dihedral angle (80, 100, 120, or 140°).

The transmembrane domain of each model was embedded in an *E. coli* inner membrane model and solvated in the CHARMM-GUI membrane builder as described for the RaptorX model. Systems were energy minimised in 5000 steps using steepest descent algorithm, and subsequently, equilibrated using the protocol described above.

Duplicates of each equilibrated system were generated. The first replicate of each system was simulated for 250 ns at 303.15 K. The topologies of the second replicates were modified to add elastic network bonds that would bring the periplasmic and transmembrane domains together. Elastic bands (of length 1 nm and strength 100 kJ mol$^{-1}$ nm$^{-2}$) were generated between the alpha carbons of 10 residue pairs, identified through coevolution analysis, to be more than 50% likely to be proximal to one another (*Table 4*).

These four replicates were simulated for 100 ns with the elastic network bonds applied. A subsequent 100 ns simulation was undertaken on each system with the network removed.

## Full OafB model: AlphaFold

For comparison to the hybrid models built above, the OafB amino acid sequence was submitted to AlphaFold (*Jumper et al., 2021*) to generate a structure prediction for the full protein. The RCSB PDB Pairwise Structure Alignment web service (*RCSB PDB, 2021*) was used to compare our structures to

**Table 5.** Position restraints used in the equilibration and simulation of the OafB-acetyl CoA systems.

| Equilibration phase | Position restraint / kJ mol$^{-1}$ nm$^{-2}$ | | | | |
|---|---|---|---|---|---|
| | Protein backbone | Protein sidechains | Lipid head groups | Dihedrals | Acetyl-coA |
| NVT1 | 4000 | 2000 | 1000 | 1000 | 4000 |
| NVT2 | 2000 | 1000 | 400 | 400 | 4000 |
| NPT1 | 1000 | 500 | 400 | 200 | 4000 |
| NPT2 | 500 | 200 | 200 | 200 | 3000 |
| NPT3 | 200 | 50 | 40 | 100 | 3000 |
| NPT4 | 50 | - | - | - | 2000 |
| MD1 | - | - | - | - | 1000 |
| MD2 | - | - | - | - | 500 |
| MD3 | - | - | - | - | - |

the AlphaFold prediction. Residues 1–327 of the RaptorX model, residues 328–406 of OafB80, and residues 407–640 of the periplasmic domain crystal structure (PDB ID: 6SE1) were aligned to the equivalent residues in the AlphaFold model using the flexible jFATCAT algorithm.

### Acetyl coenzyme A

One molecule of acetyl-CoA was generated in CHARMM-GUI's Ligand Reader and Modeler (*Kim et al., 2017*). Using VMD as a visual tool to guide positioning, the acetyl group of acetyl-coA was placed beneath the gap between TMHs 9 and 10, and the N-terminus of the OafB$_{100}$ protein. The protein was embedded in the *E. coli* inner membrane model in CHARMM-GUI and solvated in 0.15 M KCl. The system was energy minimised using the steepest descent algorithm, and subsequently, equilibrated using the protocol described above.

A short SMD simulation was used to insert acetyl-CoA into the pore (delimited by TMHs 1, 2, and 9, and the periplasmic loops between helices 5 and 6, and 3 and 4). Pulling at a rate of 0.5 nm ns$^{-1}$ and with a harmonic force of 1000 kJ mol$^{-1}$ nm$^{-2}$, the sulphur of the acetyl-CoA molecule was pulled upwards, towards the centre of mass of the alpha carbons of residues E243, D126, and E189 (all in loops at the periplasmic surface of the transmembrane domain, surrounding the pore) until the z-separation of the two groups was 0 (7.5 ns).

Three variations on the final frame of the SMD were generated in VMD by rotating and translating acetyl-coA within this pore to use as starting conformations for equilibrium MD. These systems were energy minimised using the steepest descent algorithm to resolve steric clashes, and equilibrated as described above, with additional position restraints (*Table 5*) on the acetyl-coA molecule. Each system was subjected to three consecutive MD simulations of 20 ns at 303.15 K, with decreasing position restraint strength on acetyl-coA to allow the protein to relax around the substrate.

### LPS in the periplasmic leaflet of the inner membrane

One molecule of *Salmonella spp*. LPS was generated using the CHARMM-GUI LPS modeller (O9-4 core with one O-antigen unit). Using VMD as a tool to guide positioning, two POPE and two POPG molecules were removed from the periplasmic leaflet of the OafB$_{100}$ system and the LPS lipid A moiety inserted in their place. Eight water molecules were replaced by potassium ions to neutralise the additional negative charge of the LPS. The system was energy minimised in 5000 steps via the steepest descent algorithm to resolve steric clashes. The system was equilibrated in six phases, as described for the other atomistic systems.

Steered MD was used to guide the rhamnose moiety of the LPS towards the Serine 430 residue (two replicates, one with a pull rate of 0.5 nm ns$^{-1}$, the other with a pull rate of 1 nm ns$^{-1}$; both with a pull force of 500 kJ mol$^{-1}$). The final frame of each of these trajectories (13.1 ns and 6.5 ns, respectively) was used as the starting conformation for three consecutive MD simulations at 303 K, of 20 ns each, with varying position restraints on the protein backbone (250, 0, 0 kJ mol$^{-1}$) and LPS molecule (1000, 500, 0 kJ mol$^{-1}$). Hydrogen bond analysis was undertaken in VMD (distance cut-off = 3 Å, angle cut-off = 20°) to assess the interactions between the LPS and the SGNH domain.

### DFT calculations

The electrostatic potentials of the transmembrane domain of OafB (residues 1–370) and the acetyl-CoA molecule were calculated in a vacuum using the linear-scaling density functional theory package, ONETEP (*Skylaris et al., 2005*; *Prentice et al., 2020*), using the PBE exchange-correlation functional (*Perdew et al., 1996*), augmented with Grimme's D2 dispersion correction (*Grimme, 2006*). Open boundary conditions *via* real-space solution of the electrostatics were used in a simulation cell with dimensions 9 nm × 7 nm × 8 nm. Norm-conserving pseudopotentials were used for the core electrons, and the psinc basis set, equivalent to a plane wave basis set with a kinetic energy cut-off of 800 eV, was employed. 8.0 Bohr localisation radii were used for the nonorthogonal generalised Wannier functions. The EDA implemented in ONETEP (*Phipps et al., 2015*; *Phipps et al., 2016*) was used to calculate the intermolecular interaction energy between acetyl-CoA and the AT3 domain and to extract its contributing components and to visualise charge transfer interactions.

### Acknowledgements

The authors acknowledge access to the following High Performance Computing resources: Iridis 5 at the University of Southampton and the JADE Tier 2 facility (EPSRC grant no. EP/T022205/1) to which access was granted via HECBioSim, the UK High-End Computing Consortium for Biomolecular Simulation (EPSRC grant no. EP/R029407/1). The authors also acknowledge the C Skylaris group at the University of Southampton for their help with ONETEP. KEN was supported by a Ph.D. Studentship from the Engineering and Physical Sciences Research Council (Project Number: 2446840); SNT was supported by a Ph.D. studentship from the Biotechnology and Biological Sciences Research Council White Rose Doctoral Training Program (BB/M011151/1), 'Mechanistic Biology and its Strategic Application'. SLM acknowledges the support of the Federation of European Biochemical Societies (FEBS) through a long-term fellowship. SK is supported by an EPSRC established Career Fellowship (EPSRC grant no. EP/V030779/1). We thank Alex Bateman for useful discussions and initial advice with using RaptorX.

## Additional information

### Funding

| Funder | Grant reference number | Author |
|---|---|---|
| Engineering and Physical Sciences Research Council | PhD Studentship Project Number: 2446840 | Kahlan E Newman |
| Biotechnology and Biological Sciences Research Council | White Rose DTP BB/ M011151/1 | Sarah N Tindall |
| Engineering and Physical Sciences Research Council | established Career Fellowship EP/V030779/1 | Syma Khalid |
| Federation of European Biochemical Societies | Long term fellowship | Sophie L Mader |
| Engineering and Physical Sciences Research Council | HECBioSim EP/R029407/1 | Syma Khalid |
| Engineering and Physical Sciences Research Council | High Performance Computing resource EP/ T022205/1 | Syma Khalid |

The funders had no role in study design, data collection and interpretation, or the decision to submit the work for publication.

### Author contributions

Kahlan E Newman, Formal analysis, Investigation, Methodology, Writing – original draft, Writing – review and editing; Sarah N Tindall, Formal analysis, Investigation, Methodology, Writing – original draft; Sophie L Mader, Investigation, Writing – review and editing; Syma Khalid, Gavin H Thomas, Conceptualization, Supervision, Investigation, Methodology, Writing – original draft, Writing – review and editing; Marjan W Van Der Woude, Conceptualization, Supervision, Investigation, Writing – original draft, Writing – review and editing

### Author ORCIDs

Kahlan E Newman ⓘ http://orcid.org/0000-0002-2974-528X
Sarah N Tindall ⓘ http://orcid.org/0000-0003-0292-1033
Sophie L Mader ⓘ http://orcid.org/0000-0002-3011-3319
Syma Khalid ⓘ http://orcid.org/0000-0002-3694-5044
Gavin H Thomas ⓘ http://orcid.org/0000-0002-9763-1313
Marjan W Van Der Woude ⓘ http://orcid.org/0000-0002-0446-8829

### Decision letter and Author response

Decision letter https://doi.org/10.7554/eLife.81547.sa1
Author response https://doi.org/10.7554/eLife.81547.sa2

## Additional files

### Supplementary files
• MDAR checklist

### Data availability
The trajectories generated and the run input files necessary to repeat the simulations presented here are openly available in Zenodo at https://doi.org/10.5281/zenodo.6834637.

The following dataset was generated:

| Author(s) | Year | Dataset title | Dataset URL | Database and Identifier |
|---|---|---|---|---|
| Newman KE, Tindall SN, Mader SL, Khalid S, Thomas GH, van der Woude MW | 2022 | Simulations trajectories generated and run input files | https://doi.org/10.5281/zenodo.6834637 | Zenodo, 10.5281/zenodo.6834637 |

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

## Appendix 1

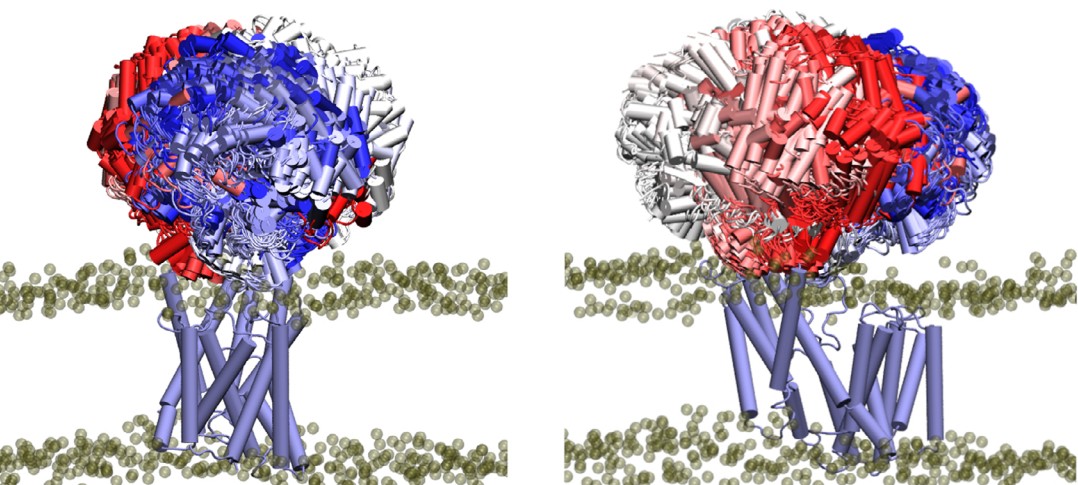

**Appendix 1—figure 1.** The SGNH domain of OafB exhibits a wide range of orientations relative to the transmembrane domain. The trajectory has been fitted (rotation and translation) to the initial position of the transmembrane domain, and the positions of the SGNH domain from every fifth frame of a single trajectory (OafB$_{100}$ without elastic network) overlaid. SGNH domain coloured by timestamp, with red = 0 ns, and blue = 250 ns.

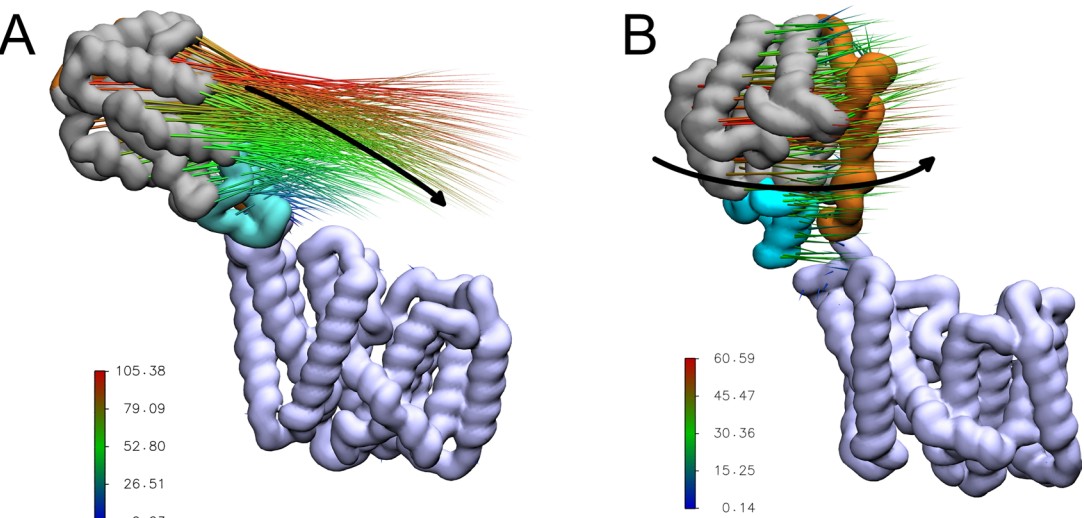

**Appendix 1—figure 2.** Porcupine plots of the first two major motions of OafB in unrestrained equilibrium molecular dynamics simulations, identified via principal component analysis.
Magnitude and direction of the motion shown with the coloured spikes: blue indicates smaller motions, red indicates larger motions. Values on colour bars indicate the magnitude of the motions in Angstroms. (**A**) A pedal bin lid-type motion is observed. The protein opens and closes, hinged at the linker (residues 370–380). (**B**) The periplasmic domain is able to rotate freely about the linker region.

