## [Editor Report]

By integrating a range of computational techniques, the authors made an important contribution by generating a structural model for the AT3 domain, which is predicted to adopt a new fold. The key features of the structural model are consistent with the activity of the enzyme as an acyltransferase, with a transmembrane channel that can accommodate an acyl-CoA donor, and an outer cavity formed with a second domain that can accommodate a nascent LPS molecule as substrate. Overall, the study is valuable as it will help stimulate specific experimental analyses that will further evaluate and improve the model for better mechanistic understanding of this class of enzymes.

---

## [Decision Letter]

**Decision letter after peer review:**

Thank you for submitting your article "A novel fold for acyltransferase-3 (AT3) proteins provides a framework for transmembrane acyl-group transfer" for consideration by *eLife*. Your article has been reviewed by 2 peer reviewers, one of whom is a member of Our Board of Reviewing Editors, and the evaluation has been overseen by Volker Dötsch as the Senior Editor. The following individual involved in review of your submission has agreed to reveal their identity: Josh V Vermaas (Reviewer #2).

Essential revisions:

1) Clarity of the presentation can be improved, and some results (e.g., open/close mechanism) can be made more robust with alternative simulation or analysis.

2) Interaction energies reported need to be placed into the proper context, so that the mechanistic implications can be made clear.

For details see the individual comments of the reviewers.

*Reviewer #1 (Recommendations for the authors):*

The combination of a broad range of computational models in the study is appropriate and powerful. The only comment/question I have concerns the quantum mechanical calculations, which revealed a very large (-1035.943 kcal/mol) interaction energy. While I understand that this represents interaction energy rather than binding (free) energy, and that there are several charged groups involved, the magnitude is rather large. Is this due in part to the fact that calculations were conducted in vacuum?

*Reviewer #2 (Recommendations for the authors):*

I would have benefitted from a reaction diagram detailing the chemistry this enzyme is thought to facilitate. The broad *eLife* readership is unlikely to know what exactly MurNAc is, or what part gets acetylated. I think, based on the extensive introduction, that the best guess right now is that this enzyme, or related enzymes, add the acetyl group onto the glucosamine nitrogen (?). These are thought to be cytoplasmic reactions, however, so I am very confused as to where this fits into polysaccharide synthesis overall.

Roughly how big is the dataset that is helping RaptorX with coevolutionary analysis? For fun, I ran this through alphafold, and indeed, the only pdb templates AlphaFold picks up on are for the soluble domain. This suggests to me that RaptorX and AlphaFold are basing the structure almost exclusively based on sequence alignments. How many sequences are close enough that the black box structural methods use them as inputs?

I would love to see some context around the quantum calculations. -1000kcal/mol sounds impressive, but is completely irrelevant to the actual binding affinity, and the way the discussion is phrased would be highly misleading to a non-computational audience.

The SI should have the sequence alignment. You clearly have computed one to know what the offset is between the current protein and other points of comparison listed in the text, but for the reader this is hard to grok the first time through, since we are naturally far less familiar with the protein sequences than you are.

Figure S2C is… unfortunate? I know that this is a VMD default visualization using Timeline, but I think we can all agree that it isn't the most professional looking representation for the data. It also argues against the highlighting in S2B, since in S2C I count 13 helical regions, not 11.

If you do an analysis that you comment on in the main text, the results should be shown. From page 15: "Principal component analysis (PCA) of the backbone atoms was used to extract the first 2 major motions (accounting for 77% or more of the variance) of the protein in the four equilibrium replicates." What is the sampled spread along these PCA dimensions? Are these even meaningful? In short simulations, this protein cannot really change its conformation, so all the PCA is measuring are the jiggles around the initial structure. The animations, while visually appealing, don't actually quantify the motions claimed in the paper. If this were framed in a more speculative fashion, I think it would be more appropriate, since the simulations are just nowhere close to exhaustive enough to prove or disprove this hypothesis.

Page 16, where can I see the occupancies, other than the selected occupancies enumerated in the text? It is not hard to tabulate this similar to what is done in Table 2.

---

## [Author Response]

Reviewer #1 (Recommendations for the authors):The combination of a broad range of computational models in the study is appropriate and powerful. The only comment/question I have concerns the quantum mechanical calculations, which revealed a very large (-1035.943 kcal/mol) interaction energy. While I understand that this represents interaction energy rather than binding (free) energy, and that there are several charged groups involved, the magnitude is rather large. Is this due in part to the fact that calculations were conducted in vacuum?

The reviewer is correct that this is, in part, due to the fact the calculation was conducted in vacuum. A clarifying statement has been added in the text (third paragraph of the subsection titled “The AT3 domain facilitates presentation of acyl-CoA molecules to the periplasm”) to reflect that this interaction energy is not a binding free energy, and that the magnitude is of less importance than the fact is energetically favourable (indicated by the negative sign).

Reviewer #2 (Recommendations for the authors):I would have benefitted from a reaction diagram detailing the chemistry this enzyme is thought to facilitate.

We have added a reaction diagram for OafB as a new Figure 1A, and in the Introduction have clarified the OafB relevant O-antigen is and the O-acetylation reaction.

The broad eLife readership is unlikely to know what exactly MurNAc is, or what part gets acetylated. I think, based on the extensive introduction, that the best guess right now is that this enzyme, or related enzymes, add the acetyl group onto the glucosamine nitrogen (?).

We thank the reviewer for the detailed interest. We now explicitly include in the introduction, and reiterate this as relevant, that the proteins mediate O-acetylation. The reviewer references MurNAc, a component of peptidoglycan which is O-acetylated by OatA, another bacterial AT3 protein that is mentioned in the Introduction. The model system and focus of this study however is the AT3 protein OafB, not OatA. OafB O-acetylates the O-antigen of the LPS. To enhance clarity further, as outlined in response to the first query, we have added a reaction diagram of the OafB mediated reaction (new Figure 1A ), and explained the O-antigen context of rhamnose acceptor more clearly in the Introduction. We have emphasised that the acylation acceptor molecules for AT3 protein mediated reactions are diverse.

These are thought to be cytoplasmic reactions, however, so I am very confused as to where this fits into polysaccharide synthesis overall.

The reviewer is correct that many acetylation reactions are cytoplasmic. We have added a sentence in the introduction acknowledging this, (and mention Lipid A acylation since it too is part of the LPS). Importantly, we then contrast that with the reactions mediated by AT3 proteins where the body of evidence supports our working hypothesis that these reactions occur extra-cytoplasmically. Our manuscript under consideration here further strengthens this hypothesis by providing a plausible answer to the problem of a cytoplasmic acetyl group donor for an extra-cytoplasmic reaction.

Roughly how big is the dataset that is helping RaptorX with coevolutionary analysis?

An alignment of about 10,000 sequences was used.

For fun, I ran this through alphafold, and indeed, the only pdb templates AlphaFold picks up on are for the soluble domain. This suggests to me that RaptorX and AlphaFold are basing the structure almost exclusively based on sequence alignments. How many sequences are close enough that the black box structural methods use them as inputs?

Alphafold will use tens of thousands of sequences of related proteins to create matrix of co-evolution co-evolved amino acids to calculate the predicted structure, as outlined in the review Skolnick et al., J. Chem. Inf. Model. 2021, 61, 10, 4827–4831. Since our initial submission, Alphafold has gained a lot of attention, has been used many times in the literature, and has gone through multiple iterations. It is now a well-established and widely accepted tool to predict protein structure.

I would love to see some context around the quantum calculations. -1000kcal/mol sounds impressive, but is completely irrelevant to the actual binding affinity, and the way the discussion is phrased would be highly misleading to a non-computational audience.

A clarifying statement has been added in the text (third paragraph of the subsection titled “The AT3 domain facilitates presentation of acyl-CoA molecules to the periplasm”) to reflect that this interaction energy is not a binding free energy, and that the magnitude is of less importance than the fact it is energetically favourable (indicated by the negative sign).

The SI should have the sequence alignment. You clearly have computed one to know what the offset is between the current protein and other points of comparison listed in the text, but for the reader this is hard to grok the first time through, since we are naturally far less familiar with the protein sequences than you are.

A sequence alignment has been added as Figure 4-supplement figure 1. Note that this is a smaller alignment than that used in Pearson et al., (2020), only including AT3 modelled in the paper, but in fact this helps identify other conserved motifs possibly involved in the acyl-CoA pore. This is referenced and discussed in subsection titled “The AT3 domain from OafB has a cavity lined with essential residues forming a putative acetyl-CoA binding site”.

Figure S2C is… unfortunate? I know that this is a VMD default visualization using Timeline, but I think we can all agree that it isn't the most professional looking representation for the data.

This refers to what is now Figure 3- supplement figure 1C. We agree it is perhaps not the prettiest way to represent the data. However, this representation does effectively convey the secondary structural features as a function of time, and is widely used and accepted in the field. Articles published in journals such as Molecular BioSystems (DOI: 10.1039/C4MB00212A), Biomolecules (DOI: 10.3390/biom3010168), Immunologic Research (DOI: 10.1007/s12026-020-09130-y), and Nature Scientific Reports (DOI: 10.1038/srep41087) use this presentation, and as such we do not feel it necessary to change this figure.

It also argues against the highlighting in S2B, since in S2C I count 13 helical regions, not 11.

The reviewer is correct that there are 13 helices present in the transmembrane domain. However, only 11 of these are transmembrane helices, as depicted in Figure 1D. A clarifying sentence has been added to the caption of Figure 3- supplement figure 1C to reflect this: “13 helices are maintained across the simulation, 11 of which are transmembrane.”

If you do an analysis that you comment on in the main text, the results should be shown. From page 15: "Principal component analysis (PCA) of the backbone atoms was used to extract the first 2 major motions (accounting for 77% or more of the variance) of the protein in the four equilibrium replicates." What is the sampled spread along these PCA dimensions?

The percentage variance explained by each eigenvector has been tabulated and added to the manuscript (new Table 1).

Are these even meaningful? In short simulations, this protein cannot really change its conformation, so all the PCA is measuring are the jiggles around the initial structure.

Whilst we agree with the reviewer that the structured regions (AT3 and SGNH domains) are unlikely to change their conformations over the short timescales sampled, here we are referring to the motion about a flexible linker which itself does not have well-defined secondary structure. This linker allows the SGNH domain to explore a large volume in the periplasm, even in the short simulations. We have added a new figure, presented as Appendix 1-Figure 1 (since we do not think we can link figures to videos). Here, every 5th frame of a simulation has been overlaid to display the different relative orientations of the SGNH and transmembrane domains. We extended these simulations to 500 ns each, and found the first two major motions to be the same as those identified in the 250 ns simulations.

The animations, while visually appealing, don't actually quantify the motions claimed in the paper. If this were framed in a more speculative fashion, I think it would be more appropriate, since the simulations are just nowhere close to exhaustive enough to prove or disprove this hypothesis.

The videos show a qualitative view of the first two major motions – a pedal-bin lid type motion is shown in Video 1, and a rotation about the linker is shown in Video 2. To strengthen the explanation of these two motions we have included a figure with the porcupine plots (Appendix 1- figure 2). The two most extreme conformations observed in the trajectories along eigenvectors 1 and 2 were selected, and the transmembrane domain of each conformation superimposed. Porcupine plots were generated to show the direction of motion of the periplasmic domain α carbons between the two conformations along each of the selected eigenvectors. The spines indicate the magnitude and direction of these motions, and match those described in the text and shown in the animations.

Page 16, where can I see the occupancies, other than the selected occupancies enumerated in the text? It is not hard to tabulate this similar to what is done in Table 2.

A table containing these data has been added (new Table 2).